# Wound Dressing with Electrospun Core-Shell Nanofibers: From Material Selection to Synthesis

**DOI:** 10.3390/polym16172526

**Published:** 2024-09-05

**Authors:** Nariman Rajabifar, Amir Rostami, Shahnoosh Afshar, Pezhman Mosallanezhad, Payam Zarrintaj, Mohsen Shahrousvand, Hossein Nazockdast

**Affiliations:** 1Department of Polymer Engineering and Color Technology, Amirkabir University of Technology (Tehran Polytechnic), Tehran P.O. Box 15875-4413, Iran; nariman.rf@aut.ac.ir (N.R.); p.mosallanezhad@aut.ac.ir (P.M.); 2Department of Chemical Engineering, Persian Gulf University, Bushehr P.O. Box 75169-13817, Iran; 3Department of Polymer Engineering, Islamic Azad University-Mahshahr Campus, Mahshahr P.O. Box 63511-41111, Iran; shahnooshafshar@gmail.com; 4Department of Biomedical and Pharmaceutical Sciences, University of Montana, Missoula, MT 59812, USA; 5Caspian Faculty of Engineering, College of Engineering, University of Tehran, Rasht P.O. Box 43841-119, Iran; m.shahrousvand@ut.ac.ir

**Keywords:** core-shell structure, wound healing, antibacterial nanomaterials, functional nanofiber electrospinning, drug delivery, nanofiber scaffolds

## Abstract

Skin, the largest organ of the human body, accounts for protecting against external injuries and pathogens. Despite possessing inherent self-regeneration capabilities, the repair of skin lesions is a complex and time-consuming process yet vital to preserving its critical physiological functions. The dominant treatment involves the application of a dressing to protect the wound, mitigate the risk of infection, and decrease the likelihood of secondary injuries. Pursuing solutions for accelerating wound healing has resulted in groundbreaking advancements in materials science, from hydrogels and hydrocolloids to foams and micro-/nanofibers. Noting the convenience and flexibility in design, nanofibers merit a high surface-area-to-volume ratio, controlled release of therapeutics, mimicking of the extracellular matrix, and excellent mechanical properties. Core-shell nanofibers bring even further prospects to the realm of wound dressings upon separate compartments with independent functionality, adapted release profiles of bioactive agents, and better moisture management. In this review, we highlight core-shell nanofibers for wound dressing applications featuring a survey on common materials and synthesis methods. Our discussion embodies the wound healing process, optimal wound dressing characteristics, the current organic and inorganic material repertoire for multifunctional core-shell nanofibers, and common techniques to fabricate proper coaxial structures. We also provide an overview of antibacterial nanomaterials with an emphasis on their crystalline structures, properties, and functions. We conclude with an outlook for the potential offered by core-shell nanofibers toward a more advanced design for effective wound healing.

## 1. Introduction

Ever since the earliest written records, the selection of materials for wound dressing has remained of paramount importance to cover and protect wounds from infection [1,2,3]. Mindful of this practice, subsequent efforts have been targeted at developing more functionalized materials assisting the healing process, which consists of hemostasis, inflammation, proliferation, and remodeling as its four main stages [4,5,6]. The evolution of wound dressings has thus seen a transition from simple materials to bioactive constructs, disclosing natural substances like oils and plant extracts as inadequately practical [7,8,9,10]. Wound dressings now aim not only to cover injuries but also to actively modulate the healing process by delivering drugs and needed remedies. The current material repertoire comprises a variety of alternatives to meet such demand, including hydrogels, alginates, and foams [11]. These dressings are developed to retain a moist environment, manage exudate, provide a barrier against pathogens, and deliver medicine to the wound site. Nevertheless, challenges still remain in achieving the ideal balance of biocompatibility, mechanical strength, and controlled release of therapeutics.

In the pursuit of advanced materials, core-shell nanofibers have been proposed as a propitious resolution due to their special structure [12,13,14]. Consisting of a core material encased within a shell, these structures at the nanoscale dimension offer spectacular interactions with the wound bed, therefore administering the therapeutic agents more effectively. Such enhancement in the functionality lies in the high surface-area-to-volume ratio (s/v) at a very low dimension [15]. Thanks to the compilation of a binary structure, i.e., core and shell components, each element can be entitled to a specific property. For wound dressing, it is common to design a core loaded with medicines, such as antibiotics and anti-inflammatory drugs, while the shell supports a protective barrier and controls the release of the core contents. The potential of dressings made from core-shell nanofibers can be further stretched to provide a conducive environment for wound healing and then gradually dissolve, reducing the need for frequent dressing changes and minimizing patient discomfort.

Although a variety of techniques, namely coaxial electrospinning, emulsion electrospinning, and microfluidic spinning, ease the development of core-shell nanofibers, implementing such complex structures for wound dressings requires firstly an understanding of the synthesis methods, along with materials’ properties and their interplay interactions [16]. For instance, coaxial electrospinning grants precise control over the core-shell morphology yet possibly fails to meet the viscosity requirements of certain bioactive agents. Obtaining a uniform drug distribution within the core is also not attainable in emulsion electrospinning despite being versatile. Concerning this issue, biocompatibility, non-toxicity, and the ability to maintain the structural integrity of the materials are essential [17,18,19]. Noting the association of the dressing with skin lesions, imparting an antibacterial property seems inevitably necessary without compromising the core-shell structure or manipulating the drug loading efficiency and controlled release kinetics [20,21]. The chosen method for wound dressing application is therefore expected to produce fibers with consistent morphology, including size and surface characteristics conducive to wound healing.

Despite the alluring prospects given by core-shell nanofibers, reports on these materials are scarce. In this review, we will begin by providing an overview of nanofibers with various structures and discuss their synthesis methods, architecture, and applications. We will then highlight the common materials used in core-shell nanofibers for wound dressings, such as thermoplastics, natural polymers, and antibacterial agents, accompanied by a survey of the recent reports on each element. Finally, we will discuss relevant fabrication methods to develop core-shell nanofibers. We hope this review can underscore the promise of core-shell nanofibers for wound dressing applications, specifically their design flexibility and modulation, and motivate researchers to investigate coaxial structures for biomedical applications.

## 2. Wound Dressing: An Emerging Field of Study

Each year, many people experience various forms of skin damage resulting from flames or accidents with the potential of significant impairment due to the high cost of treatment, and in severe cases, the risk of death. To rectify these challenges, wound dressings are increasingly emphasized in the development of efficient medical treatments. As shown in Figure 1A, over 2500 research articles were published in 2023 focusing on wound dressing applications. As of the date of reporting this review (June 2024), more than half of last year’s papers conform to the same field of interest, supporting its importance in academia. In fact, one reason for spurring interest in developing wound dressings is that the traditional wound care methods fail to address the complex needs of distinct wound types, giving rise to prolonged healing times, infections, and suboptimal recovery. Moreover, the increasing prevalence of chronic wounds, such as diabetic ulcers, pressure ulcers, and venous leg ulcers, has further fueled the surge in wound dressing research. Chronic wounds are often associated with impaired healing processes, necessitating advanced wound care products that can facilitate tissue regeneration, provide antimicrobial protection, and maintain a moist healing environment. As an innovative solution, the interest in core-shell nanofibers was ignited in late 2003 by the pioneer report of Sun et al. [22] on coaxial electrospinning. With the same trend as wound dressing, enthusiasm for core-shell nanofibers has thereafter initiated in academia, as shown in Figure 1B.

To date, several review papers have been published with an emphasis on core-shell nanofibers and wound dressings, each employing distinct approaches to present previous findings, ranging from lab-scale experiments to full-scale studies. These reviews cover various aspects, from the electrospinning method to biopolymer-derived materials used for fabricating core-shell nanofibers, with many concentrating on wound dressing and other biomedical applications. Few reviews also explore nanostructured materials for the same purposes yet are not fully dedicated to the core-shell structure. To clarify the approach of the current paper, Table 1 compiles recently published review articles in this field, providing a comprehensive overview of the literature. In contrast to other studies, our review attempts to bridge the existing gap between the materials used in core-shell nanofibers and the most effective methods to develop such complex architecture for wound dressing applications.

### Ideal Wound Dressing

In contrast to bandages that immobilize or compress an injured area to aid in healing or control bleeding, dressings are in direct contact with a wound to promote the healing process much more effectively and protect it from further harm. The selection of a suitable wound dressing is dependent upon the type of wound, location, and depth, as well as the amount of exudate, presence of infection, and degree of wound adhesion. Conventional dressings such as cotton bandages are known to absorb a significant amount of moisture from the wound, leading to drying of the wound surface and deceleration of the healing process, which causes discomfort during dressing removal. Modern dressings, on the other hand, are designed to maintain an optimal moist environment conducive to wound healing, i.e., renewing the skin without any eschars. By preserving adequate moisture levels, dressings not only enhance the healing rate but minimize pain associated with dressing changes. 

The extensive use of polymers in wound care represents a significant improvement over traditional methods, offering tailored solutions that cater to the specific needs of different wound types and conditions. Wet dressings based on polymeric materials, therefore, are deemed to be the promising types of dressings. An ideal wound dressing encompasses important characteristics, including regulation of moisture around the wound, removal of excess exudates, protection against infections and microorganisms, high gas permeability, reduction of wound surface necrosis, provision of mechanical protection, ease of application and removal, biocompatibility, cost-effectiveness, and alleviation of wound pain, as shown in Figure 2.

## 3. Nanofibers

From the technical perspective, nanofibers are circular cross-section threads with diameters in the range of nanometers (usually below 100 nm), exhibiting properties that distinguish them from their bulk counterparts [34,35,36,37]. Due to high surface area along with chemical and mechanical properties, core-shell nanofibers represent a unique class of materials with scientific and technological benefits. The importance of nanofibers is underscored by their extensive range of applications across multiple fields, including but not limited to tissue engineering scaffolds, wound dressings, drug delivery systems, wearable electronics, and water filtration [31,38,39,40]. Advancements in nanofiber fabrication methods have led to design flexibility and ease of material choice, with electrospinning providing significant advantages in simplicity, versatility, and scalability [41,42]. As shown in Figure 3, nanofibers are typically classified into six groups concerning their structural design. In the following section, we will highlight common morphologies, applications, and synthesis methods of nanofibers.

### 3.1. Core-Shell Nanofibers

Nanofibers with a core-shell structure are designed to encapsulate one material (inner part or core) with another substance (shell). This architecture facilitates the combination of diverse properties, resulting in such performance that is usually impracticable by solid nanofibers. For instance, the core can be composed of a polymer with high stiffness and the shell provides biocompatible characteristics [43]. One notable property of this structure is the ability to release therapeutic agents in a controlled manner [44,45,46]. While the shell protects the inner material from degradation and regulates the diffusion rate, the core encapsulates drugs or other bioactive molecules, making the core-shell nanofibers particularly suitable for drug delivery applications. In this regard, it has been shown that a Fluorouracil (5-FU)-loaded core can be delivered and emitted from a polyvinylpyrrolidone (PVP) shell over a specified period, enhancing treatment efficacy and reducing side effects [47]. This report demonstrates that a composition of polymers in the shell manipulates the drug release regarding the hydrophilicity and degradation rate of the used materials. The effective control of 5-FU using core-shell nanofibers as a customized cancer treatment protocol accounts for the importance of this work as the mentioned medicine suffers from a short half-life.

Drug-loaded core-shell nanofibers have also been utilized for wound dressings [48,49]. As shown by a work under the supervision of Prof. Zomorodian [50], a core-shell nanofiber with a core made of polyvinyl alcohol (PVA) and chitosan (blended), and polyethylene oxide (PEO) and gelatin shell supports well-tolerated, affordable dressing for cutaneous leishmaniasis by providing compelling drug delivery as well as mechanical strength. Results from tensile testing revealed high strength and modulus of the drug-loaded core-shell nanofibers in comparison with unloaded nanofibers. This behavior was attributed to the plasticizing effect of the drug due to its resembling molecules with core polymer, reducing the intermolecular forces between the polymer chains and therefore more flexible structure. MTT assays, a cell viability test, indicated no cytotoxicity towards fibroblast cells, confirming the biocompatibility of the nanofibers. The core-shell nanofibers exhibited a controlled release profile, releasing 84% of Glucantime within the first nine hours.

The core-shell nanofibers are synthesized through various methods, with coaxial electrospinning being the most prevalent [51,52]. Specifically, in this method, two different polymer solutions are simultaneously fed through a coaxial needle, forming a compound jet under a high-voltage electric field that elongates and solidifies into nanofibers with a core-shell morphology. A more detailed explanation of coaxial electrospinning will be provided in the processing section.

### 3.2. Porous Nanofibers

Porous nanofibers are characterized by their internal structure where numerous cavities or pores of varying sizes exist [53,54,55]. The high s/v stands out as the most notable characteristic of this structure, stemming from both the nanoscale dimension of the fibers and the additional surface area provided by the pores [56,57]. Required for adsorption, catalysts, and sensing applications, this property is advantageous for extensive surface interaction. Researchers have recently highlighted that porous nanofibers host a higher number of active sites compared to their non-porous counterparts, thereby increasing the reaction rate and efficiency [58,59]. Another significant feature of porous nanofibers is elevated permeability by intrinsic porous structure that facilitates the transport of molecules and ions through the fibers, as reported by a work under the supervision of Prof. Menkhaus [60]. This investigation states that porous lignin carbon nanofibers greatly eliminate contaminants due to their ability to trap particles within their pores while allowing water molecules to pass through. The combination of high surface area and porosity has not only been exposed for industrial end uses but wound dressing is also in demand for such practicality. In a work conducted at Aarhus University [61], it was reported that a functionalized porous nanofiber with tannic acid showed faster wound closure compared to uncoated as well as solid nanofibers, which corresponded to the ability to maintain a moist environment and supporting cell growth derived by the high porosity of the material. This structural design also brought about more epidermal tissue regeneration, reduced inflammation, and increased collagen deposition, thus better healing dynamics.

To fabricate porous nanofibers, several methods are utilized, including phase separation, self-assembly, and electrospinning, with the latter being the most frequently cited in the literature [62,63]. By changing the parameters of the electrospinning process, such as polymer concentration, solvent volatility, and applied voltage, the phase separation within the jet is induced, leading to the formation of pores. The porous nanofibers are engineered to present various mechanical properties depending on the material composition and the porosity level. It is well known that increasing porosity tends to decrease mechanical strength; however, selecting appropriate polymer matrices or incorporating reinforcing agents can mitigate such poor attributes [64,65].

**Figure 3 polymers-16-02526-f003:**
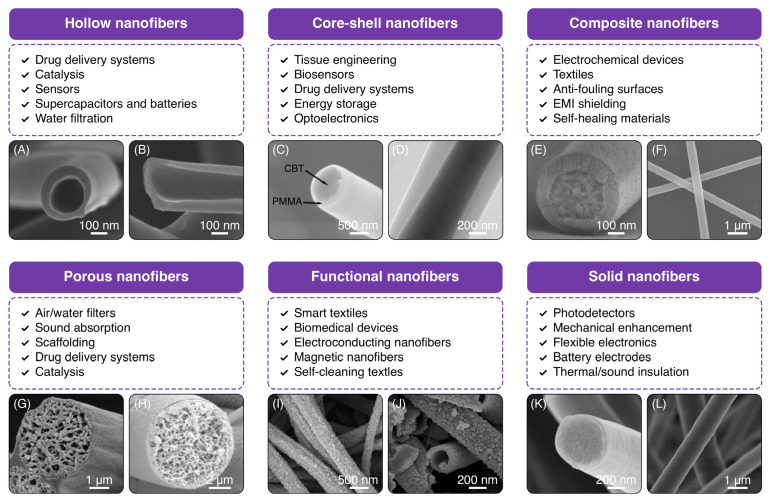
SEM images of common structures in nanofibers. (**A**,**B**) Hollow carbon nanofibers (Reproduced from ref. [66] with permission from Nature). (**C**) CBT nanofiber encapsulated with PMMA, (**D**) TEM image of a core-shell nanofiber made of pyridine-TPU as the shell and TPU as the core (Reproduced from refs. [67,68] with permission from IntechOpen and eXPRESS, respectively). (**E**,**F**) Modified polystyrene nanofibers with solvents (Reproduced from ref. [69] with permission from American Chemical Society). (**G**,**H**) Porous nanofibers made of polystyrene after etching (Reproduced from refs. [70,71] with permission from Springer and American Chemical Society, respectively). (**I**,**J**) AgNPs coated on polyphenols/polystyrene nanofibers (Reproduced from refs. [72,73] with permission from Elsevier). (**K**,**L**) PVDF nanofibers (Reproduced from ref. [74] with permission from Springer).

### 3.3. Composite Nanofibers

Composite nanofibers have left a mark on materials science with their prominent advances, combining multiple components to achieve superior properties that surpass those of the individual constituents [75,76,77]. The synthesis of composite nanofibers involves blending a polymer solution with reinforcing agents before the fabrication process. As with other nanofiber morphologies, the electrospinning method is particularly suited for creating composite nanofibers since it allows for the homogeneous dispersion of reinforcing materials within the polymer matrix [78,79,80]. Compared to other structures as well as pristine polymers, composite nanofibers depict enhanced tensile strength and Young’s modulus owing to the intrinsic higher stiffness. An investigation carried out by Wang et al. [81] has shown that the incorporation of graphene oxide (GO) into the PVA matrix gave rise to an almost 175% increased tensile strength upon the addition of 8 mg GO. This team concluded that such enhancement is empowered by the high aspect ratio and exceptional mechanical strength of GO, reflected by the interaction between GO and the matrix.

The multifunctional properties of composite nanofibers render them highly versatile for a diverse range of applications, from biomedical and energy storage devices to electronics, high-performance textiles, and filtration [82,83]. Pursuing the biomedical application of composite nanofibers, bioactive nanoparticles including hydroxyapatite (HA) or silver nanoparticles are plausible to impart antibacterial properties and enhance biocompatibility, which is crucial for wound healing and implantable devices. Wang et al. [84] have reported a core-shell nanofiber made of chitosan and gelatin for wound dressing applications featuring antibacterial properties. This work showed the dual-functional role of nano HAP for drug encapsulation and nanofiber reinforcement. Briefly, a tetracycline hydrochloride aqueous solution as an antibiotic was dropwise added to a suspension of nano HAP, followed by mixing it with chitosan and gelatin solution for nanofiber electrospinning. Results on the prepared composite nanofibers reported maintaining an optimal moisture level for wound healing without causing excessive dehydration or accumulation of wound exudates, along with demonstrating the sustained release of the antibacterial agent after nine days. The inclusion of nano HAP also improved the elastic modulus of the nanofibers to nearly 113% yet flexible enough for protecting the wounds.

### 3.4. Functionalized Nanofibers

Surface modification stands as a feasible approach to improve the overall performance of materials [85]. The term “functionalized” refers to chemically or physically modified nanofibers to introduce specific functional groups or moieties, thereby imparting unique properties absent in the pristine fibers. Accomplished through various techniques including surface coating, grafting, or incorporation of nanoparticles into a matrix, these modifications are often carried out in post-fabrication steps. Nevertheless, in some cases, the functional components can be introduced during the electrospinning process itself [86,87,88,89,90,91,92]. The primary objective of functionalization is to tailor nanofiber properties for specific applications, thus enhancing their versatility. The improvement of mechanical properties, however, remains the foremost benefit of this process.

A work carried out by Zhu et al. [93] has highlighted the improvement of the mechanical behavior of poly(ester-urethane) urea (PEUU) nanofibers grafting with peptides. Attributed to the strong tethering of peptides to the PEUU molecular chains, their results revealed an almost 2.5-fold higher tensile strength and stiffness when peptides are covalently grafted onto the nanofibers. They further investigated the effect of such grafting on the bioactivity of the nanofibers. It was shown that grafted PEUU-peptide nanofibers had high cell adhesion and proliferation pertaining to the biocompatibility of the samples. In fact, by incorporating peptides, the surface tended to be more conducive to cellular activities, making these nanofibers suitable for wound healing and vascular grafts.

### 3.5. Hollow Nanofibers

The hollow nanofibers are characterized by their morphological feature where a central cavity is surrounded by a solid shell [94,95,96]. Due to the deliberate hollow design, hollow nanofibers are lightweight such that they bring promises to a wide range of applications when weight reduction is crucial without compromising strength. Moreover, this structure has been referred to as an ideal design for membranes and filtration [97,98,99].

Hollow nanofibers are produced through advanced fabrication methods such as coaxial electrospinning, template-assisted techniques, and self-assembly processes. As a common synthesis method, coaxial electrospinning involves the extrusion of two polymer solutions through a coaxial needle, creating a composite fiber that has a core-shell configuration [100]. The outer polymer solution forms the shell, whereas the inner solution constitutes the core and then is removed, resulting in a hollow fiber morphology. Alternatively, hollow nanofibers can be fabricated using sacrificial templates.

### 3.6. Solid Nanofibers

Nanofibers with a uniform structure, often simply referred to as solid nanofibers, are a fundamental class of nanomaterials characterized by their continuous, non-porous, and homogeneous construction. The term “solid” signifies that the nanofibers are composed of a single material throughout their cross-section, without any internal pores/cavities or secondary phases. Despite being solid, the surface area offered by this morphology is quite high due to the small dimension of the fibers, ensuring adequate surface interaction with other components. Beyond this advantage, the primary attribute of solid nanofibers lies in their favorable tensile strength as well as flexibility owing to their continuous and defect-free structure. PVA nanofibers, for instance, have shown tensile strengths ranging from 20 to 100 MPa, depending on the processing conditions and fiber alignment [101]. This significant improvement in mechanical properties makes solid nanofibers suitable for applications in protective textiles, tissue engineering scaffolds, and structural composites.

In the field of biomedical engineering, solid nanofibers are used extensively as scaffolds for tissue regeneration. Their high surface area promotes cell attachment and proliferation, while their mechanical properties provide the necessary support for tissue growth. Solid nanofibers also find applications in filtration systems due to their ability to form dense and uniform mats that can effectively capture particles and contaminants.

## 4. Materials Selection

The integrated functionality of the core-shell nanofibers is determined by the properties of each material. Upon tailoring the composition of both core and shell elements, the desired performance and versatility are achieved for a wide range of applications, particularly for biomedical devices. In a typical core-shell architecture, the core material serves as the structural foundation, providing the necessary mechanical integrity and stability to the nanofibers. The shell material, on the other hand, dictates modulating the surface characteristics and functionalities of the nanofibers. This layer is often guided toward the aim of the core-shell nanofibers for drug delivery, biodegradability, or specific biological activities.

A number of materials are inherently resistant to pathogens, obviating the requirement for an additional element [102,103]. To this end, chitosan, alginate, silk fibroin, and cellulose-based materials are all featured by antibacterial attributes due to their ability to disrupt bacterial cell membranes, maintain hydrated environments, and chelate essential metal ions. However, several drawbacks including low mechanical strength, poor processability to obtain a consistent nanofiber, and sensitivity to environmental conditions like temperature constrain their widespread use. Moreover, the scalability of these nanofibers is prone to be challenging when their expenditure is noted [104]. The overall tendency of utilizing natural polymers (Figure 4) in the core-shell nanofibers is therefore attributed to the core layer. Beyond these polymers with instinctive antibacterial properties, the addition of inorganic materials including silver nanoparticles seems essential in the shell content, rendering the nanofibers for wound dressing and tissue engineering applications. In the following section, we will lay the basis of materials selection to fabricate shell and core components of nanofibers with a focus on wound dressing applications. We will provide an overview of the recent reports of antibacterial agents used in core-shell nanofibers.

### 4.1. Materials for Core-Shell Nanofibers

As the outermost layer, the shell accounts for promoting wound healing and inhibiting bacterial growth (Figure 5). This layer is designed to depict good cell adhesion and proliferation properties, whereas the core material facilitates the controlled and sustained release of therapeutic agents. In wound dressing applications, the shell material must exhibit antibacterial properties to inhibit bacterial colonization and prevent infection at the wound site. These properties are typically achieved through the incorporation of antimicrobial agents, such as silver nanoparticles, into the shell matrix. Due to the importance of the mechanical robustness of the core-shell nanofibers, thermoplastics are mostly preferred to form the shell layers [105,106]. Synthetic polymers show sufficient mechanical strength and flexibility to conform to the contours of the wound site alongside maintaining structural integrity during use. They are also ideal candidates for the induction of porosity to facilitate exudate absorption and cell infiltration. In core-shell nanofibers, either bio-based thermoplastics or natural polymers are used to fabricate the shell component. To gain a better understanding of the required properties for producing a core-shell nanofiber, we highlight the most common materials.

#### 4.1.1. Synthetic Polymers

Polymers are preferred as the core or shell element of core-shell nanofibers due to their versatility, tunable molecular weight and architecture, and desirable interfacial properties. Among various polymers, polyesters are usually utilized as they boast controlled degradation, biocompatibility, as well as economical advantages. Polyesters are composed of ester monomers, consisting of a carboxylic acid group and an alcohol moiety linked by an ester bond (-COO-), posing various configurations from linear to branched [107,108]. Excellent mechanical strength, chemical resistance, and biocompatibility render polyesters ideal candidates for fabricating nanofibers with enhanced properties through structural variations. Opportunities to develop deformation-resistant prototypes with ease of fabrication have garnered attention to exploring poly(lactic acid) (PLA) and poly(ε-caprolactone) (PCL) as synthetic ester-based polymers. Each of these polymers can be functionalized or blended with other materials to suit nanofiber production.

*Poly(lactic acid) (PLA):* PLA is a biodegradable thermoplastic derived from renewable resources such as corn starch, sugarcane, or potato [109,110]. Consisting of lactic acid monomers via ester linkages, PLA has three stereoisomers including L-PLA, D-PLA, and the racemic mixture DL-PLA with different crystallinity. Compared to traditional thermoplastics, PLA merits mechanical strength, ease of processability, and transparency. The degradation rate of PLA can be tuned by adjusting its molecular weight and non-amorphous content, providing flexibility in designing structures with desired degradation profiles to match specific biomedical needs [111,112].

In a work conducted by Fang et al. [113], PLA was used as the shell material due to its good biodegradability, and poly(γ-glutamic acid) (γ-PGA) as the core using the coaxial electrospinning method to create core-shell nanofibers (Figure 6A). In vitro studies of PLA/γ-PGA nanofibers demonstrated good biocompatibility, with cell viability over 110% compared to controls. In vivo experiments on mice showed that the nanofiber membrane enhanced wound healing considerably. Treated wounds exhibited over 90% re-epithelialization within 14 days, compared to only 25% in untreated controls. The nanofiber structure, with its high surface area and porosity, likely facilitated cell attachment and proliferation, contributing to accelerated wound closure. An investigation under the supervision of Prof. Yildirim showed the potential of encapsulated PLA with gelation for wound dressing applications [114]. The PLA was designed to provide a hydrophobic yet mechanically robust core, complementing the hydrophilic and bioadhesive characteristics of gelatin (Gel). Upon optimization of the materials ratio as well as the electrospinning parameters, the PLA/gelatin nanofiber with a diameter of 100 nm indicated a dual-drug delivery capability and excellent cell proliferation properties. The optimized nanofibers mimicked the extracellular matrix, promoting cell adhesion and growth, which are crucial for efficient wound healing and tissue regeneration. In another research carried out at Nantong University, a three-layer core-shell nanofiber was developed using PLA as the core containing polydopamine (PDA) and polypyrrole (PPy) as the coat on the PLA [115]. This strategy was considered due to the low hydrophilicity of PLA for wound repair, aiming to improve the reactive oxygen species (ROS) scavenging capacity of the nanofiber and antibacterial properties. The results showed successful modification of the hydrophilicity of PLA upon decreasing the contact angles from 110.8° (neat PLA) to 30.9° (PPy/PDA/PLLA). The nanofibers also exhibited strong antibacterial properties, reducing *Escherichia coli* (*E. coli*) and *Staphylococcus aureus* (*S. aureus*) viability under near-infrared light (Figure 6B). Additionally, the PPy/PDA/PLLA nanofibers enhanced wound healing in a rat model, accelerating hemostasis and promoting angiogenesis and epidermal recovery. The material’s near-infrared photothermal response (around 47.1 °C) contributed to its bactericidal effect and enhanced wound healing, suggesting its potential as a multifunctional wound dressing.

*Poly(vinyl alcohol) (PVA):* PVA is a biocompatible, water-soluble polymer produced by the hydrolysis of polyvinyl acetate [116,117]. The solubility and physical properties of PVA are contingent on the degree of polymerization and hydrolysis, indicating the extent to which the acetate groups are converted to hydroxyl groups. These groups contribute to the adhesion properties, rendering PVA able to form strong bonds with other materials. The non-toxic nature and swelling in aqueous environments further make PVA suitable for biomedical applications, such as drug delivery systems, wound dressings, and tissue engineering scaffolds, as it provides gradual dissolution in the body [118,119].

In a work conducted at Shiraz University, researchers combined PVA with chitosan as the core material and PVP/maltodextrin (MD) as the shell [120]. The strategy involved coaxial electrospinning to produce nanofibers with a core-shell structure. The core contained PVA/chitosan with essential oils (*Satureja mutica* or *Oliveria decumbens*) to impart antimicrobial and antioxidant properties, while the shell consisted of PVP/MD to enhance mechanical strength and control the release of bioactive agents. The resultant PVA-based nanofibrous scaffolds demonstrated a uniform, beadless morphology with fiber diameters of 210 ± 50 nm for the unloaded scaffold and 250 ± 45 nm and 225 ± 46 nm for scaffolds loaded with *S. mutica* and *O. decumbens* essential oils, respectively (Figure 7A). The mechanical testing showed a tensile strength of 8.92 ± 0.1 MPa for the scaffold without essential oils, which was within the suitable range for wound dressings. This team found that the antimicrobial activities against *E. coli*, *S. aureus*, and Candida species improved with 10% essential oils, making them promising candidates for wound dressing applications by providing both antimicrobial and mechanical strength.

A recent report led by Prof. Osfouri at the Persian Gulf University highlighted the potential of core-shell nanofibers based on PVA and sodium alginate (SA) for wound dressing applications [121]. In this study, the researchers employed an electrospinning technique to fabricate nanofibers. PVA was blended with SA to form the core element, along with using chitosan as the shell layer with an aim for better mechanical strength and drug release control. They utilized dexpanthenol as a wound-healing agent in the core. The shell of the nanofibers, particularly with 1% chitosan, controlled the gradual release of dexpanthenol, following the Fickian diffusion mechanism as modeled by the Korsmeyer-Peppas equation. This report demonstrated that PVA significantly enhanced the wettability and swelling capacity of the wound dressing, with the swelling ratio increasing from 226.3% for pure PVA to 282% for PVA/SA nanofibers. The nanofibers also exhibited a beadless structure with average diameters ranging from 100 to 130 nm depending on the shell composition. MTT assays and cell culture studies revealed that the dexpanthenol-loaded PVA/SA/chitosan nanofibers were non-toxic to fibroblast cells and promoted cell attachment and proliferation. A work conducted by Maleki et al. [122] showed the antibacterial activity of a core-shell nanofiber made of coated PVA with silver nanoparticles (AgNPs) and PLA. Involving coaxial electrospinning for developing this nanofiber, PVA was selected as the core owing to its high water solubility, biocompatibility, and ability to host antibacterial agents. Initially, silver nitrate (AgNO_3_) was incorporated into the PVA solution, followed by adding hydrazine hydrate in a post-electrospinning treatment to reduce AgNO_3_ to AgNPs. This method resulted in the controlled release of antibacterial agents due to the encapsulation of AgNPs within the PVA core and on the surface of the fibers. For wound dressing application, the PVA/Ag-PLA nanofibers demonstrated efficient antibacterial activity as the presence of AgNPs led to increased inhibition zones for *E. coli* and *S. aureus*. The fibers exhibited uniform morphology with diameters ranging from 300 to 600 nm.

**Figure 7 polymers-16-02526-f007:**
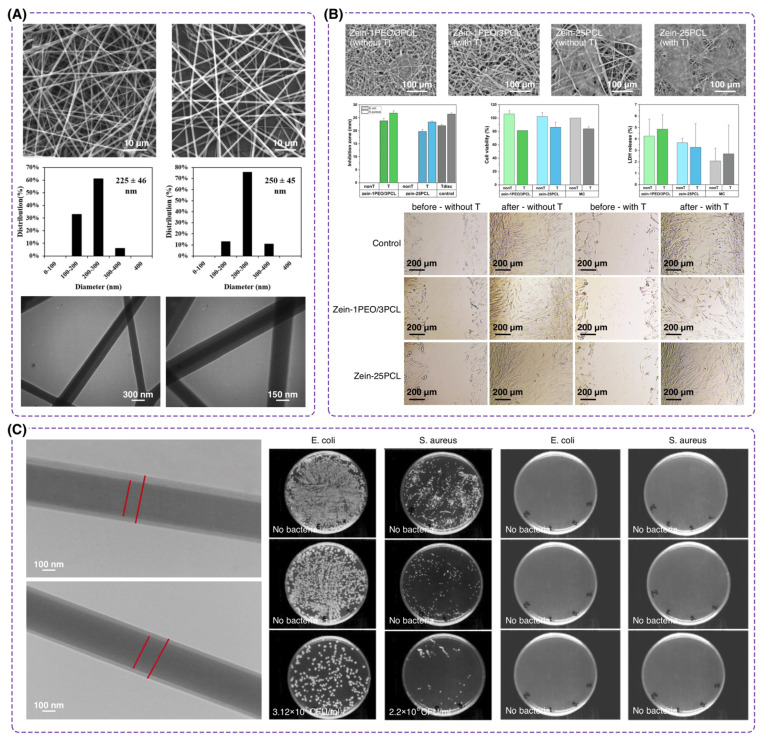
Core-shell nanofibers for wound dressing. (**A**) SEM images of chitosan/PVA nanofibers with 10% *O. decumbens* and *S. mutica* are shown alongside their diameter range. TEM images exhibit the formation of core-shell nanofiber structure. (Reproduced from ref. [120] with permission from Elsevier). (**B**) SEM images of fibroblast cell adhesion on zein-PCL nanofibers. Bar charts illustrate the antimicrobial and cell viability results of zein-PCL nanofibers. Images of wound scratch assay with zein-PCL core-shell nanofibers are provided. (Reproduced from ref. [123] with permission from Elsevier). (**C**) TEM images of chitosan/PCL core-shell nanofibers with their antibacterial activity. (Reproduced from ref. [124] with permission from Elsevier).

*Poly(ε-caprolactone) (PCL)*: Composed of ε-caprolactone monomer, PCL is marked as an easy-to-process biodegradable polymer due to its flexibility derived by low glass transition temperature ($T_{g}$) around −60 °C [125]. Tailoring the properties of PCL for distinct applications, it is synthesized through the ring-opening polymerization of ε-caprolactone, typically catalyzed by stannous octoate, allowing for precise control over the molecular weight and molecular weight distribution [126,127]. Within the body, PCL is degraded through hydrolysis of its ester bonds, eventually metabolizing into non-toxic products such as 6-hydroxycaproic acid. Previous reports have demonstrated modulating the degradation rate of PCL by adjusting its molecular weight and crystallinity, providing convenience in design and processing [128]. In terms of chemical properties, PCL is also known as a polar and hydrophobic polymer that remains stable in aqueous environments. Although several works have addressed the utilization of PCL in the core of a core-shell structure, PCL is well suited to form the shell layer due to its compatibility with a wide range of substances, along with mechanical properties [124,129,130,131,132]. The high extension and tensile strength of PCL contribute to the durability and flexibility of the core-shell structures, thereby providing mechanical resistance when high stress is imposed. Such properties enable PCL to create dressings that conform well to the wound surface, providing optimal coverage and protection [133,134]. The hydrophobic nature of PCL permits maintaining a dry environment on the outer surface of the dressing, while its degradation products are non-toxic and do not cause adverse reactions.

In a recent work conducted at the University of Copenhagen, a core-shell nanofiber based on PCL and zein (a hydrophobic protein) was developed [123]. Zein core was initially mixed with an antimicrobial agent, tetracycline hydrochloride (TCH), and then a layer of PCL was electrospun to form the shell using glacial acetic acid as a solvent. The zein-PCL fibers demonstrated reduced diameter and hydrophobicity, along with sustained release of TCH and fibroblast attachment as favorable parameters for wound closure (Figure 7B). The zein-25PCL (25% PCL) fibers showed enhanced mechanical properties with water stability and significant tensile strength, maintaining their structure in a hydrated environment. This stability, along with efficient fibroblast attachment and antimicrobial activity, suggests that zein-PCL core-shell nanofibers are promising candidates for developing advanced biodegradable wound dressings with prolonged drug release capabilities, potentially reducing the frequency of dressing change. Another study led by Ghazalian showed the encapsulation of chitosan with PCL to deliver the same antibacterial agent as the former group [124]. Employing a coaxial electrospinning method to fabricate core-shell nanofibers, they preferred the combination of chitosan for its biocompatibility and antibacterial properties while the mechanical strength is supplied by PCL. Three different sets of core-shell nanofiber samples were tested, and the optimal composition was determined to be 2 wt% chitosan for the core and 15 wt% PCL for the shell, both in acetic acid, for desirable morphology during electrospinning. The chitosan/PCL core-shell nanofibers demonstrated a two-stage drug release profile containing an initial burst and sustained release. This behavior is notable for maintaining an antibacterial environment over extended periods, thus enhancing wound healing. The core-shell nanofibers also showed adequate antibacterial activity, completely inhibiting the growth of *E. coli* and *S. aureus* and retaining their fibrous structure during the in vitro degradation experiment (Figure 7C).

*Cellulose acetate:* Cellulose acetate is an ester derivative of purified cellulose obtained from the acetylation process, where the hydroxyl moieties are replaced with acetyl groups [135]. This process involves a strong acid, such as sulfuric acid, as a catalyst along with acetic anhydride to substitute the functional groups. The acetylation process improves its solubility, fabrication, and thermal stability. Despite native cellulose with highly crystalline content, cellulose acetate is soluble in various organic solvents due to the disruption of hydrogen bonds during the treatment. The average number of hydroxyl groups substituted per glucose unit in the cellulose chain is referred to as the degree of substitution (DS), resulting in the characteristics of cellulose acetate. At low DS content (i.e., monoacetate), cellulose acetate retains more hydroxyl groups, rendering it more hydrophilic, less thermally stable, less fragile, and more opaque. These properties conform to an inverse trend upon increasing the DS ratio. Cellulose acetate is usually utilized in the shell of the core-shell nanofibers owing to its biocompatibility, biodegradability, favorable mechanical strength, flexibility, and permeability to gas while being a barrier against pathogens [136,137].

In a work conducted at Oklahoma State University, the mechanical properties and cell viability of cellulose acetate as the shell layer were studied [138]. This group compared the properties of solid cellulose acetate nanofiber, its hollow structure, and core-shell nanofibers embodying PCL with various molecular weights as the core element. Results showed that electrospun nanofibers with the coaxial fabrication method led to higher elastic elongation and tensile strength compared to the unencapsulated structures, specifically under hydrated conditions. Although hollow cellulose acetate nanofibers exhibited nearly double the cell viability, they found that cellulose acetate/PCL nanofibers supported robust cell adhesion and spreading without any signs of cytotoxicity, demonstrating the successful encapsulation of PCL. In a very recent work from the University of Shanghai, the controlled drug release from cellulose acetate shell in a core-shell nanofiber was investigated [139]. This team developed a novel electrospinning technique, utilizing cellulose acetate as the matrix in both layers. They employed a modified coaxial electrospinning method, where the shell layer consisted of a non-spinnable solution (cellulose acetate and ZnO) that was rendered spinnable by the core solution (cellulose acetate and curcumin) through its driving force. This study showed that the shell effectively controlled the release of curcumin and Zn^2^⁺, achieving a two-day synergistic release profile. When the shell was loaded with ZnO nanoparticles, the penetration of water molecules into the core was delayed, extending the release period of curcumin and prolonged covering of the wound.

*Polyurethane (PU):* PU is a copolymer formed by the reaction of diisocyanates (hard segments) with polyols (soft segments), resulting in a polymer chain with a block alternation of dissimilar segments [140]. Depending on the molecular weight and ratio of the monomers, the microphase separation between these two segments is tailored to meet specific mechanical properties to suit a range of uses [141]. In wound dressing, PU-based core-shell structures are preferred due to their permeability to gases, allowing oxygen to reach the wound while preventing the accumulation of excess moisture, which can lead to maceration [142,143,144]. The chemical reactivity of PU, particularly the isocyanate groups, allows for various chemical modifications. These modifications can introduce functional groups that enhance the material’s properties or impart new functionalities. For instance, grafting hydrophilic polymers onto PU can improve its wettability and biocompatibility, making it more suitable for medical applications. Similarly, conjugating drugs or targeting ligands onto PU can create targeted drug delivery systems that release therapeutic agents at specific sites in the body, improving treatment efficacy and reducing side effects.

In a recent report by Movahei et al. [142], core-shell nanofibers based on PU, starch, and HA were developed using the coaxial electrospinning method for wound dressing application (Figure 8). PU was selected to form the core layer referring to its good mechanical properties, and the shell was made of starch and HA for better biocompatibility and hydrophilicity. The MTT assay indicated cell viability rates above 80% after seven days of incubation for all scaffolds, confirming their non-cytotoxic nature. Microscopic images revealed that cell proliferation on PU/starch and PU/starch/HA nanofibers was significantly higher than on pure PU scaffolds after one and four days. Moreover, the quantitative results from in vitro tests indicated that the cell viability increased from day one to day seven. The PU/starch/HA nanofibers indicated an absorbance of 1.75 at 490 nm on day seven compared to 1.25 for PU/starch and 0.95 for PU scaffolds, demonstrating enhanced cell viability and proliferation. Using a rat model, this team investigated an in vivo wound healing efficacy as well. Wounds treated with PU/starch/HA scaffolds demonstrated the highest healing rate, with approximately 80% wound closure by day 14 compared to 65% for PU/starch. Histological analysis revealed that the PU/starch/HA treated wounds showed a well-developed dermis, the presence of hair follicles, and sebaceous glands, indicating superior tissue regeneration. The number of inflammatory cells was significantly lower in the PU/starch/HA group, highlighting its effectiveness in reducing inflammation and promoting faster healing.

*Polyacrylonitrile (PAN):* PAN is a synthetic polymer consisting of acrylonitrile monomers, characterized by a linear chain configuration with nitrile groups (-C≡N) attached to each carbon backbone [145,146]. This structure imparts PAN with high tensile strength, excellent thermal stability, and remarkable chemical resistance properties which are critically advantageous when PAN is employed in core-shell nanofibers [147,148]. The high tensile strength of PAN is typically attributed to the strong intermolecular interactions between the nitrile groups, which facilitate extensive hydrogen bonding and dipole-dipole interactions. As a shell material, PAN can protect the core element from thermal degradation owing to the nitrile groups with a high melting point required to break their bond, thereby extending the overall lifespan and performance of the core-shell structure [149,150,151]. Chemical resistance is a further advantageous property of PAN, making it resistant to a wide range of solvents, acids, and bases. This resistance is primarily due to the strong carbon-nitrogen triple bonds within the nitrile groups, which are less susceptible to chemical attack. In addition to its mechanical and chemical properties, PAN exhibits good barrier properties against gases and liquids [152,153]. The dense molecular structure of PAN creates a barrier that is impermeable to gases and liquids, making it suitable for applications requiring effective containment or separation, such as membranes. The impermeable nature of PAN also enhances its functionality as a shell material, preventing the diffusion of unwanted substances into the core, thereby preserving the core’s integrity and functionality.

In a recent report from Qingdao University, researchers used PAN to serve as a structural backbone in the fabrication of nanofibers owing to its excellent mechanical properties [154], chemical stability, and ability to form continuous fibers through electrospinning (Figure 9). They employed two approaches to improve the functionality of PAN, including the doping method and the secondary growth strategy. The first method involved embedding the ZIF-8@gentamicin within the PAN/gelatin nanofibers during the electrospinning process. The ZIF-8 stands for zeolitic imidazolate framework-8, a type of metal-organic framework (MOF) that features a porous structure with a high surface area for drug delivery purposes. Gentamicin is also an antibiotic, loaded in ZIF-8, to achieve a controlled and sustained release profile. In the secondary growth method, ZIF-8 nanoparticles were first grown on the surface of the PAN/gelatin nanofibers and subsequently loaded with gentamicin. The secondary growth method resulted in higher drug loading capacity and a more controlled drug release, attributed to the surface-grown ZIF-8 nanoparticles providing more binding sites and reducing the initial burst release of the drug. The PAN-based composite nanofibers, particularly those prepared by the secondary growth method, exhibited superior antibacterial properties and enhanced wound healing capabilities. These nanofibers effectively reduced wound healing time from 21 days to 16 days in a mouse model of wound infection. The combination of PAN with gelatin and the coating of ZIF-8@gentamicin created a synergistic effect, providing a conducive environment for cell growth while simultaneously combating bacterial infections.

#### 4.1.2. Natural Polymers

As an alternative to synthetic polymers, naturally derived biopolymers including chitosan, silk fibroin, collagen, gelatin, and alginate are likely to be used to fabricate the core layer, sometimes the shell, since they boast antibacterial properties. Such desirable feature is attributed to their ability to disrupt bacterial cell membranes and chelate essential metals, thereby inhibiting bacterial growth [155,156,157]. These biopolymers not only provide a biocompatible and biodegradable platform but also promote cell adhesion, proliferation, and wound healing. Their structural similarity to the extracellular matrix promotes cellular interactions and tissue integration, which is important for effective wound healing.

*Chitosan:* Chitosan is a linear polysaccharide derived from the exoskeletons of crustaceans, e.g., shrimp and crabs, containing deacetylated and acetylated monomers [158,159,160]. The primary source of chitosan is chitin, known as the second most abundant natural polymer after cellulose, featuring amino groups. These amino groups render chitosan able to form gels and films alongside reactions with a variety of substances through hydrogen bonding. In other words, because of the protonation of amino groups, chitosan is dissolved in acidic solutions, providing easy processing and manipulation into films and fibers. The biocompatibility and antibacterial properties of chitosan make it an ideal material for wound dressings as it prohibits infections and adverse immune responses, therefore faster healing is attainable [161,162]. In fact, chitosan has shown broad-spectrum antimicrobial activity against bacteria, fungi, and viruses, primarily due to its polycationic nature, which disrupts microbial cell membranes. Besides the molecular weight, it has been reported that the degree of deacetylation of chitosan manipulates its mechanical properties, resulting in the required flexibility and strength to support tissue formation for scaffolds [163,164].

In a work conducted by Keirouz et al. [165], a core-shell nanofiber based on nylon-6 and chitosan was developed to prohibit mesh-associated surgical site infections (Figure 10A). They utilized chitosan due to its polycationic nature, which allows it to permeabilize bacterial cell walls and therefore osmotic imbalances and inhibit bacterial growth. Using the coaxial electrospinning method, they considered a core-shell structure with nylon-6 at the core and a chitosan/PEO shell for antimicrobial action. The addition of PEO improved the electrospinnability of chitosan by reducing the repulsion between the polymer chains, leading to the production of uniform fibers. This study is significant due to the inhibition of the growth of both Gram-positive and Gram-negative bacteria, as well as controlled drug release. The controlled drug release enhances the antimicrobial efficacy over a prolonged period, which is vital for ideal wound dressing and infection control. In another report by Hasanbegloo et al. [166], chitosan was used as the core containing paclitaxel-loaded liposomes to enhance the functional properties of the nanofibers, providing a sustained and controlled release of the drug (Figure 10B). The core was supported by PCL, using the coaxial electrospinning technique. They showed that chitosan/PCL core-shell nanofibers improved wound healing due to the antimicrobial activity of loaded chitosan. In vitro cytotoxicity studies showed that liposome-loaded core-shell nanofibers resulted in more than 85% cancer cell death after 168 h, compared to 57% for simple chitosan nanofibers. Additionally, in vivo results indicated a reduction in tumor weight from 1.35 g to 0.65 g using this functionalized core-shell nanofiber, demonstrating the potential for effective localized drug delivery and enhanced wound healing.

*Silk fibroin:* Silk fibroin, a natural protein derived from the silk of the Bombyx mori silkworm, is renowned for its biocompatibility, biodegradability, and favorable mechanical properties [168,169,170]. The main structure of silk fibroin consists of a sequence of amino acids, predominantly glycine, alanine, and serine, which form repetitive crystalline regions (also known as β-structures sheet) interspersed with amorphous regions. The hydrophilicity of silk fibroin lies in the presence of polar amino acids. Because such ordered and disordered segments coexist, silk fibroin withstands a notable stress load [171,172]. Meanwhile, the capability of absorbing and dissipating the energy is expected regarding the presence of amorphous content. The minimal immunogenicity of silk fibroin makes it highly promising to use as the shell material in core-shell structures for wound dressing applications [173,174,175]. Resembling PCL, the biodegradability of silk fibroin allows slight degradation in the body, reducing the risk of long-term side effects and eliminating the need for surgical removal of the material. Silk fibroin is also a suitable material for wound dressing as it absorbs water, maintaining a moist environment [176,177].

In a work led by Hadisi at the University of Victoria, the core-shell nanofibers were developed forming the core element with silk fibroin and the surface layer with hyaluronic acid (HA) by employing the coaxial electrospinning technique [178]. Also, they utilized zinc oxide (ZnO) nanoparticles in the fibers to provide antibacterial properties, enhancing wound healing by preventing infections. The in vitro cytotoxicity studies demonstrated that the core-shell nanofiber containing 3 wt% ZnO (as the optimal ratio of nanoparticles) caused improved cell proliferation compared to other formulations. In vivo studies further validated the efficacy of these nanofibers, showing that the dressing led to a 55.02% wound closure after seven days, significantly higher than the 21.69% observed in control groups. Histopathological analysis revealed better epidermis regeneration, higher collagen formation, and reduced inflammatory response in wounds treated with this formulation. Another report under the supervision of Prof. Dhara leveraged emulsion electrospinning for fabricating core-shell nanofibers containing silk fibroin in the core and PCL in the shell [167]. They investigated the efficiency of high ratios of silk fibroin compared to the shell content on wound healing (Figure 10C). Their findings support that high silk fibroin content in the electrospun fibers enhanced cell adhesion and proliferation, as confirmed by in vitro studies with human placenta-derived mesenchymal stem cells. The in vivo rodent model demonstrated that the core-shell structure of silk fibroin/PCL at a 70/30 ratio significantly accelerated full-thickness wound healing, promoting hair follicle development, and reducing scar formation within 15 days, thus highlighting its potential for effective skin regeneration applications.

*Alginate:* Alginate, a polysaccharide derived from brown seaweed, is extensively utilized in biomedical and environmental applications due to its biocompatibility and unique physicochemical properties [179,180]. Alginate is composed of linear copolymers of β-D-mannuronic acid (M) and α-L-guluronic acid (G) residues, arranged in blocks of homopolymer M or G regions and heteropolymer MG regions. The relative proportion and sequence of these blocks determine the physical properties of alginate, such as its gelation behavior, mechanical strength, and solubility [181,182]. Alginate offers major applications for biomedical ends by encapsulating and controlling the release of therapeutic agents and bioactive molecules.

Norouzi et al. [183] reported a sodium alginate/PCL core-shell nanofiber using an emulsion electrospinning approach (Figure 11). Upon optimization of polymer contents and surfactant, they prepared this emulsion of sodium alginate and PCL in a stable water-in-oil media to encapsulate alginates. They found a stable emulsion for the electrospinning process is achieved using a 4% *w*/*v* concentration of sodium alginate, a 1% surfactant concentration, and a water-to-oil ratio of 0.1. They also investigated the cell viability and biocompatibility of sodium alginate/PCL core-shell nanofibers for wound dressing applications. The membranes were tested to evaluate cytotoxicity for indirect cytotoxic effects by culturing normal human dermal fibroblast cells with membrane extracts for 24 and 48 h. The results exhibited no cytotoxic effects from the extracts, proving that the sodium alginate/PCL nanofibers were biocompatible and did not harm the cells. In a work conducted at Isfahan University of Technology, researchers compared two synthesis methods of drug-loaded alginate/PEO nanofibers for wound dressing application [184]. In blended electrospinning, the first approach, sodium alginate and PEO were mixed with vitamin C, while core-shell electrospinning as the second method encapsulated vitamin C within an alginate/PEO shell. The core was formed by the addition of vitamin C to PEO. They chose the core-shell structure to enhance control over drug release and protect the encapsulated medicine from environmental degradation. The core-shell nanofibers exhibited a lower initial release rate of vitamin C compared to blended nanofibers, demonstrating their potential for sustained drug delivery. This slower release was highlighted as beneficial for wound healing owing to a prolonged therapeutic effect and reduces the frequency of dressing changes, thereby minimizing disruption to the healing process. Additionally, the cross-linking of nanofibers with glutaraldehyde vapors improved stability and controlled the degradation rate of core-shell nanofibers, further supporting their suitability for wound dressing applications.

#### 4.1.3. Antibacterial Materials

The effectiveness of nanomaterials as antibacterial agents is attributed to several parameters, starting with the surface area (Table 2). As a nature of any nano-scaled structure, antibacterial agents with a dimension of 1–100 nm (known as the standard definition of a nano-scaled material) have a large surface area relative to their volume, enhancing their interaction with bacterial cells for more effective bacterial adhesion and interaction [185,186,187]. Another critical feature of antibacterial nanomaterials is their ability to generate reactive oxygen species (ROS) [188]. Materials like titanium dioxide (${TiO}_{2}$) and ZnO can induce the formation of ROS under specific conditions, such as UV irradiation [188,189]. ROS, including hydroxyl radicals, superoxide anions, and hydrogen peroxide, are highly reactive and cause oxidative damage to bacterial cell components, including lipids, proteins, and DNA [190,191]. This oxidative stress leads to cellular dysfunction and death. Due to the higher efficiency, ability to act with different mechanisms, possible synergistic impact with other materials, and least side effects, nanomaterials with antibacterial properties are preferred to integrate with other materials. Here, we review some of the most common antibacterial nanomaterials used in core-shell nanofibers.

*Silver nanoparticles (AgNPs):* AgNPs mark the first place when it comes to selecting an antibacterial nanomaterial [192,193,194,195]. Widely recognized for their potent antibacterial activity, silver nanoparticles work by releasing silver ions, which interact with bacterial cell membranes and disrupt cellular functions, leading to cell death. However, the size of nanoparticles is contingent on antibacterial activity. Smaller nanoparticles, typically in the range of 1–20 nm, exhibit better antibacterial properties compared to larger particles due to the increased surface area [196,197]. Additionally, smaller nanoparticles can more easily penetrate bacterial cell walls and membranes, leading to more effective intracellular interactions. In a report under the supervision of Prof. Montalti, it was highlighted that AgNPs with sizes around 5 nm exhibit higher antibacterial activity against both Gram-positive and Gram-negative bacteria compared to larger particles [198]. Among various morphologies of AgNPs, such as rod-shaped, triangular, cubic nanoparticles, and spherical, the latter one is the most used due to its relatively simple synthesis and stable properties [199,200]. However, certain shapes like triangular and rod-shaped nanoparticles have shown enhanced antibacterial activity, potentially due to their increased surface area and the presence of sharp edges that can cause physical damage to bacterial membranes [201,202,203]. For example, triangular silver nanoparticles have been reported to exhibit higher antibacterial efficacy compared to spherical nanoparticles of similar size, likely due to their ability to interact more effectively with bacterial cells [204,205,206].

The crystal structure of silver nanoparticles is another critical factor that determines the antibacterial properties. Silver nanoparticles typically crystallize in a face-centered cubic (fcc) structure, which is the most stable form [207,208]. The fcc structure provides a high density of active surface sites, which enhances the release of silver ions (${Ag}^{+}$). These ions are known to be highly reactive and can interact with various bacterial cellular components, including the cell membrane, proteins, and DNA. The release of silver ions from AgNPs is a key mechanism underlying their antibacterial activity. For instance, it has been demonstrated that the antibacterial efficacy of AgNPs is directly correlated with the rate of silver ion release, which is influenced by the crystal structure and surface characteristics of the nanoparticles [209,210].

*Zinc oxide (ZnO):* ZnO nanoparticles have been widely examined for their nontoxicity and bactericidal effect [211,212,213]. Similar to AgNPs, the crystalline configuration of ZnO dominates the antibacterial activity in which the hexagonal wurtzite unit cell is widely referred to as the most effective structure against *Escherichia coli* and *Staphylococcus aureus* among spherical, and rod-like [214,215]. Nonetheless, spherical morphology is known as a common structure due to the ease of synthesis with a similar size.

ZnO typically crystallizes in the hexagonal wurtzite structure, characterized by a non-centrosymmetric arrangement of zinc and oxygen atoms. This crystal structure is promising for antibacterial activity due to its high density of active surface sites and the presence of polar surfaces that facilitate interactions with bacterial cells [216,217]. The wurtzite structure namely enhances the generation of ROS and the release of zinc ions (${Zn}^{2+}$), both of which contribute to the antibacterial action of ZnO nanoparticles. The antibacterial efficacy of ZnO is closely linked to its ability to generate ROS such as hydroxyl radicals, superoxide anions, and hydrogen peroxide under certain conditions, including UV irradiation [218,219,220,221]. The generation of ROS by ZnO nanoparticles leads to oxidative stress, causing damage to bacterial cell components such as lipids, proteins, and DNA, ultimately resulting in cell death. The disruption of bacterial membranes by ZnO nanoparticles involves direct physical interaction, resulting in increased membrane permeability and leakage of intracellular contents [222,223]. Additionally, zinc ions released from ZnO can penetrate bacterial cells and interact with various intracellular targets, inhibiting essential cellular processes such as enzyme activity and protein synthesis.

*Graphene oxide (GO):* GO nanoparticles boast high mechanical strength and versatile surface chemistry, rendering them ideal for high effectiveness in combating bacterial infections, particularly against antibiotic-resistant strains [224,225,226]. Smaller GO nanosheets, typically with lateral dimensions in the range of 20–500 nm, exhibit enhanced antibacterial properties compared to larger sheets. The increased s/v ratio of smaller GO nanoparticles promotes greater interaction with bacterial cells, supporting more effective adhesion, membrane disruption, and intracellular interactions [227].

GO is typically synthesized as two-dimensional nanosheets with a high aspect ratio [228]. These nanosheets can exhibit sharp edges and a high density of functional groups on their surface, which contribute to their antibacterial efficacy. The sharp edges of GO nanosheets can physically pierce bacterial cell membranes, leading to mechanical disruption and leakage of intracellular contents. This physical damage is further exacerbated by the oxidative stress induced by the functional groups present on the GO surface.

The degree of oxidation of graphene oxide is another critical factor determining its antibacterial properties [229,230]. GO is characterized by the presence of various oxygen-containing functional groups, such as hydroxyl, epoxy, and carboxyl groups, on its basal planes and edges. These functional groups contribute to the hydrophilicity, dispersibility, and reactivity of GO nanoparticles. The degree of oxidation, often quantified as the oxygen-to-carbon (O/C) ratio, affects the density and distribution of these functional groups. GO with a higher degree of oxidation generally exhibits greater antibacterial activity due to the increased generation of ROS and enhanced interactions with bacterial membranes.

*Copper oxide (CuO):* CuO nanoparticles are another antibacterial nanomaterial that has been cited in the literature [231,232]. With a range of 10–50 nm, the rod-shaped CuO nanoparticles have shown significant antibacterial effects against *E. coli* and *S. aureus* compared to spherical nanoparticles of similar size [233,234]. In terms of the crystal structure, copper oxides typically exist in two oxidation states: cupric oxide (CuO) and cuprous oxide (Cu_2_O). CuO, with a monoclinic crystal structure, is the most studied form for antibacterial applications due to its stability and high surface reactivity [235,236]. The monoclinic structure of CuO provides a high density of active surface sites and releases copper ions (${Cu}^{2+}$), conforming to the same mechanism in bacterial suppression as discussed for ZnO.

*Titanium dioxide (TiO_2_):* TiO_2_ nanoparticles have drawn considerable attention in the field of antibacterial research due to their unique properties, including their high stability, low toxicity, and strong photocatalytic activity, rendering them highly effective against a wide range of bacterial pathogens [237,238,239]. Various morphologies have been synthesized and studied for their antibacterial efficacy of TiO_2_, including spherical, rod-like, and anatase nanocrystals. Despite the ease of synthesis and stable properties of spherical TiO_2_ nanoparticles, rod-like and anatase nanocrystals have been reported to exhibit enhanced antibacterial activity, potentially due to their higher aspect ratios and larger specific surface areas. In this regard, a report led by Kyzas has shown that anatase TiO_2_ nanocrystals have greater antibacterial effects against *Escherichia coli* and *Staphylococcus aureus* compared to spherical nanoparticles of similar size [240].

Besides the morphology, the crystal structure of TiO_2_ is also an important factor in determining its antibacterial properties [241]. TiO_2_ exists in three primary crystalline forms: anatase, rutile, and brookite. Among these, the anatase phase is the most studied and utilized for antibacterial applications due to its superior photocatalytic activity. The anatase structure is indeed a tetragonal crystal system characterized by a higher degree of crystallinity and a larger band gap compared to the rutile and brookite phases [242]. This higher band gap (typically 3.2 eV) makes anatase TiO_2_ more effective in generating ROS under UV irradiation, therefore more potent in battling against bacteria.

## 5. Processing Methods

Similar to the material selection, the implementation of a processing method for core-shell nanofibers is essential since it leaves a significant impact on the morphology, structure, and functional properties of the nanofibers (Figure 12). To maintain the distinct core-shell architecture during the process, the selected materials need to be combined in a manner that preserves their characteristics and ensures the formation of a well-defined core-shell structure. The intermixing of the core and shell materials and the formation of defects/irregularities within the nanofibers are unfavorable as they directly affect the quality and functionality of nanofibers. In the following section, we will highlight the most common techniques to prepare core-shell nanofibers. An overview of the merits and demerits of these methods is tabulated in Table 3.

### 5.1. Coaxial Electrospinning

Coaxial electrospinning entails the simultaneous electrospinning of two distinct polymer solutions, resulting in the formation of nanofibers with a core material encapsulated within a shell [244,245,246,247,248]. This technique is known as a versatile approach to fabricating nanofibers, permitting the incorporation of diverse materials in the core and shell components, thereby enabling the design of nanofibers with tailored properties. However, the setup and operation of coaxial electrospinning are complex compared to single-fluid electrospinning, necessitating precise control over multiple parameters to achieve the desired fiber morphology. Coupled with the requirement for specialized equipment and materials, this complexity increases the overall cost of the process [249,250]. Additionally, scaling up the coaxial electrospinning process for industrial production remains a challenge, as maintaining uniformity and consistency in fiber production at large scales is arduous.

The coaxial electrospinning setup comprises a high-voltage power supply, a coaxial spinneret, syringe pumps, and a collector. The coaxial spinneret is designed with two concentric needles: an inner needle for the core solution and an outer needle for the shell solution. In a typical process, polymer solutions are pumped through their respective needles at controlled flow rates by syringe pumps. The ratio of core to shell materials can indeed be modulated by varying the flow rates of the respective solutions, enabling precise control over the thickness and properties of the core and shell components [67,251]. Upon employing a high voltage, a compound Taylor cone is formed at the spinneret tip, and electrostatic forces elongate the polymer solutions into a jet. This jet undergoes stretching and whipping instabilities, leading to the formation of core-shell nanofibers which are later collected on a grounded or rotating collector. The efficiency of coaxial electrospinning is contingent upon the properties of the polymer solutions including the viscosity, conductivity, surface tension, applied voltage, flow rates of the core and shell solutions, and the distance between the spinneret and the collector. In addition, the selection of solvents can significantly influence the electrospinning process by modulating the evaporation rate and solidification of the fibers.

### 5.2. Emulsion Electrospinning

Another method to fabricate the core-shell nanofibers is emulsion electrospinning, which leverages the principles of electrospinning by involving the use of a high-voltage electric field to draw fine fibers from a polymer solution [252,253]. The processing of this method begins with an emulsion comprising an aqueous phase and an oil phase stabilized by surfactants. This system allows for the encapsulation of hydrophilic substances in the aqueous phase and hydrophobic substances in the oil phase, making it a versatile method for producing multifunctional nanofibers.

The emulsion is loaded into a syringe, which is connected to the needle. The syringe pump controls the flow rate of the emulsion. When a high voltage is applied to the needle, the emulsion at the needle tip forms a Taylor cone and is ejected as a jet. As the jet travels towards the grounded collector, the solvents evaporate, and the emulsion droplets elongate and solidify into nanofibers. The core-shell structure is achieved because the phases in the emulsion tend to separate under the influence of the electric field and solvent evaporation dynamics.

One of the primary advantages of emulsion electrospinning is the ability to encapsulate and protect sensitive active ingredients within the core, such as drugs, enzymes, or proteins, helpful in biomedical applications where controlled and sustained release of therapeutic agents is desired [254,255,256]. In addition, this method is particularly preferred to develop multifunctional fibers by incorporating different substances into the core and/or shell. Substitution of organic solvents with water is furthermore a win for the emulsion electrospinning for more environmentally friendly processes, reducing the reliance on harmful and expensive solvents. Despite its potential, emulsion electrospinning faces several challenges that need to be addressed to fully realize its capabilities [257,258]. One major challenge is the reproducibility of the fiber structures and properties. Variations in emulsion preparation and electrospinning conditions can lead to significant differences in the final product, which can be problematic for applications requiring high precision and consistency. The long-term stability of the encapsulated substances has also remained a concern, as the protective shell might degrade over time or under certain conditions, leading to premature release or deactivation of the active ingredients. While the implementation of water gives the edge to the sustainability of this technique, the environmental impact of the surfactants and other additives is still a concern, which can be toxic or difficult to remove from the final product. The quality of the core-shell nanofibers obtained from the emulsion electrospinning process depends on the concentration and type of polymers, the emulsion composition, and the electrospinning conditions [259,260]. Although efficient encapsulation and uniform fiber morphology seem tricky in this method, a number of studies have reported the viability of doing so upon optimizing these parameters.

**Table 3 polymers-16-02526-t003:** Comparing the pros and cons of common electrospinning methods for core-shell nanofibers.

Method	Pros	Cons
Coaxial electrospinning	- Flexible in material combinations - Ideal for drug delivery application - High encapsulation efficiency of bioactive agents - Continuous production - Precise control over fiber diameter and shell thickness	- Complex setup - Safety concerns due to high voltage power supply - Not compatible with all materials - Requiring the use of solvents - Possible issues with nozzle clogging
Emulsion electrospinning	- Straightforward setup and process - Effective encapsulation of hydrophilic and hydrophobic materials - Compatible with a wide range of polymers and bioactive agents - Controlled release by tuning emulsion composition - Using aqueous emulsions	- Challenging to maintain a stable emulsion - Less precise control over core-shell structure - Limited scalability - Less uniform nanofibers - Possible poor fiber properties due to residual emulsifiers
Phase separation	- Straightforward setup and process - Producing fibers with high porosity - Compatible with different polymers - No need for high voltage power supply - Scalable for large-scale production	- Limited control over core-shell structure - Need for organic solvents in most cases - Slow-paced process - Less structural stability in fibers - Restriction for continuous production
Template synthesis	- Accurate control over fiber dimensions - Highly uniform and consistent fibers - Compatible with a wide range of polymers - Capable of creating multi-component structures - Scalable for large-scale production with appropriate template design	- More expensive - Requiring preparation of templates - Slow-paced process - Requires removal of the template, involving additional processing steps and solvents - Restriction for long, continuous fibers
Centrifugal spinning	- High production rate compared to electrospinning methods - Relatively simple and inexpensive setup - No need for high voltage power supply - Compatible with different of polymers - More sustainable due to less solvent requirement	- Less uniform nanofibers - Offers less control over core-shell structure - Inferior mechanical properties - Rougher surface properties - Not compatible with all materials due to their viscosities - Less precise control over core-shell structure

### 5.3. Template Synthesis

Template synthesis is a common approach to producing core-shell nanofibers due to its precision and ability to develop complex nanostructures boasting certain properties [261,262,263,264]. The principle of this method involves a template material to guide the formation of the nanofibers, which can later be eliminated to leave behind the desired core-shell structure. The template synthesis typically consists of various elements, starting with a base material known as the template. Often a porous membrane or a preformed nanofiber is initially prepared. Although the desired size of the core-shell nanofibers and their morphology dictate the template, anodic aluminum oxide (AAO) membranes and polycarbonate track-etched membranes are renowned for this step. Once the template is ready, the core material is deposited within the pores or around the nanofibers of the template via chemical vapor deposition (CVD), electrochemical deposition, or solution-based processes. In the template synthesis method, a diverse range of materials such as polymers, metals, and semiconductors are utilized for core structure [265,266]. After the core material is deposited, the shell material is introduced, typically by coating the entire template-core assembly with the desired shell substance. Techniques like dip-coating, spin-coating, or additional deposition processes are employed to form the shell layer. Once the core-shell structure is formed, the template is removed through either chemical etching, dissolution, or thermal degradation to obtain the final nanofiber.

The template synthesis is preferred when advanced structural nanofibers are required, enabling a high degree of control over the dimensions (i.e., diameter, length, and wall thickness) and morphology of the nanofibers that pertain to selecting the appropriate template and deposition technique [267]. Furthermore, the independence to select materials for the core and shell layers grants freedom to develop complex, multi-material nanofibers that are difficult to achieve through other methods. Nonetheless, requiring a post-processing step to eliminate the template material without damaging the core-shell structure is a notable pitfall of this method. The template removal treatment can be a detrimental process requiring harsh chemical or thermal treatments that inevitably affect the integrity of the nanofibers, particularly if the core/shell materials need some sensitivity caution. Same as any post-processing procedure, this additional work can be also costly or inconsistent to prepare in large quantities, limiting the scalability of the template synthesis method. High adaptability to remove the template in the template synthesis technique typically translates to a high yield of nanofibers with uniform dimensions and properties.

### 5.4. Centrifugal Spinning

Centrifugal spinning, also known as force spinning, is an innovative technique for the fabrication of core-shell nanofibers [268,269,270,271,272]. This method utilizes centrifugal force to produce nanofibers from polymer solutions or melts, offering several advantages over traditional electrospinning, particularly in terms of production rate and fiber morphology. In centrifugal spinning, the polymer solution or melt is placed in a rotating spinneret, which is then spun at high speeds to generate the necessary force to eject the polymer through small orifices, forming fibers as the solvent evaporates or the melt cools. By employing coaxial spinnerets or sequential spinning processes, centrifugal spinning can be adapted to create core-shell structures.

The setup for centrifugal spinning involves a motorized spinneret capable of high rotational speeds, a reservoir for the polymers, a collection system for the fibers, and a heating system if polymer melts are used. The spinneret, often resembling a cylindrical container with small nozzles or apertures, holds the polymers. Upon rotation, the centrifugal force propels these materials through the nozzles, stretching them into fine fibers. In coaxial centrifugal spinning, a coaxial nozzle is used where two concentric spinnerets enable simultaneous extrusion of two different polymer solutions or melt, resulting in core-shell fiber formation. Alternatively, sequential spinning can be employed, where a core fiber is first produced, followed by coating it with a shell layer using a secondary spinning process.

Unlike electrospinning or template synthesis, centrifugal spinning leaps nanofiber production at a high throughput due to the absence of an electric field in the case of electrospinning, which limits the processing speed [273,274,275]. The equipment required is generally less complex and more robust than that used in electrospinning, facilitating easier scale-up for industrial applications. Also, centrifugal spinning enables the use of high-viscosity polymers as well as high-ratio polymer contents. Such possibility grants the production of thicker and more robust fibers. Furthermore, this technique is less sensitive to the electrical properties of the solution, making it suitable for a broader range of materials, including conductive and non-conductive polymers. Centrifugal spinning, however, suffers from fluctuations in the flow rate of the polymers, causing variability in fiber thickness and morphology. This non-uniform fiber diameter becomes trickier when a high production rate is required. Additionally, the rapid solvent evaporation or melt cooling can result in incomplete shell formation or structural defects in the fibers.

### 5.5. Phase Separation

Phase separation is an effective technique for fabricating core-shell nanofibers with complex internal structures [276,277,278]. This process relies on the separation of two or more polymer phases within a solution or melt, leading to the formation of distinct core and shell regions upon solidification. The ability to control the phase separation dynamics allows for precise tailoring of the nanofiber morphology, making this approach valuable for a variety of applications requiring multifunctional nanostructures.

The phase separation process begins with the preparation of a homogeneous solution or melt containing at least two polymers. As a critical point, the selected polymers need to be either immiscible or partially miscible to promote phase separation. Not only the polymers but the utilized solvent is required to dissolve the polymers to form a homogeneous solution, followed by selectively evaporating to induce phase separation. Once the polymer solution or melt is prepared, it is processed into nanofibers using techniques such as electrospinning or centrifugal spinning. Electrospinning is particularly prevalent, where the polymer mixture is fed through a needle under a high-voltage electric field, drawing fine fibers as the solvent evaporates. During this process, the differences in polymer affinities for the solvent and their molecular weights cause them to separate into distinct phases. Typically, the polymer with the higher affinity for the solvent forms the outer shell, while the other polymer aggregates to form the core. Controlling the process parameters, including solvent evaporation rate, polymer concentration, and spinning conditions, is essential to ensure a uniform core-shell nanofiber [279].

The core-shell nanofibers are more likely to be obtained with more well-defined and stable structures in this method without the need for complex coaxial nozzles or post-processing steps. The spontaneous segregation of polymers into distinct phases simplifies the fabrication process, making it more straightforward compared to other methods discussed earlier. Phase separation also offers significant control over the internal morphology of the nanofibers. By adjusting the concentration of the polymers, solvent composition, and processing conditions, it is possible to fine-tune the size and distribution of the core and shell regions. However, one significant challenge of this method is the difficulty in predicting and controlling the exact phase separation dynamics, which can be influenced by a myriad of factors including polymer-solvent interactions, ambient conditions, and processing parameters. Achieving uniform core-shell structures across large batches of fibers can be challenging, as slight variations in any of these factors can lead to inconsistencies in the final morphology.

## 6. Conclusions and Outlook

Core-shell nanofibers exemplify a rising advancement in materials science due to numerous applications leveraging such complex structures. In the field of biomedical and wound dressing, the importance of core-shell nanofibers is even more critical as several pitfalls of the current dressings can be resolved upon broadened implementation. The unique architecture of core-shell nanofibers offers a controlled and sustained release of therapeutic agents which facilitate a prolonged antimicrobial effect and promote faster wound recovery. Unlike other coverings, the core-shell nanofibers also notably improve the mechanical strength and stability of the dressings, making them more durable and versatile.

In this review, we explored the various structural designs of nanofibers, emphasizing the superiority of core-shell structure in wound dressing applications. The choice of materials and synthesis methods was discussed as critical in developing a practical wound dressing. The selection of an ideal wound dressing material hinges on several factors, including biocompatibility, affordability, non-toxicity, antimicrobial activity, and the ability to maintain a moist wound environment. By providing a versatile platform for incorporating various bioactive agents, recent surveys showed that core-shell nanofibers excel in these aspects. Thermoplastic polymers alongside natural-based materials were highlighted for their biocompatibility, biodegradability, and inherent wound-healing properties, as well as antibacterial nanomaterials.

We believe the integration of self-healing polymers, responsive hydrogels, and smart polymers into core-shell nanofibers enhances their performance even further by providing targeted, on-demand therapeutic delivery. These materials can be designed to respond to pH, temperature, moisture, and enzymes or biological molecules, thereby providing efficient controlled drug delivery. Incorporating bioactive molecules, such as growth factors and peptides, can further promote wound healing and tissue regeneration. Improving fabrication techniques seems also crucial for scalable, high-quality production. To mimic the wound site more accurately, exploring alternative methods like 3D printing and self-assembly is necessary for creating intricate geometries and multifunctional nanofibers. Beyond the material selection and methods, in vivo and clinical studies are essential to validate the safety, efficacy, and biocompatibility of these wound dressings to translate findings into practical applications. Further investigation is therefore required so as to elevate the current advancement of core-shell nanofibers toward an ideal wound dressing.

## Figures and Tables

**Figure 1 polymers-16-02526-f001:**
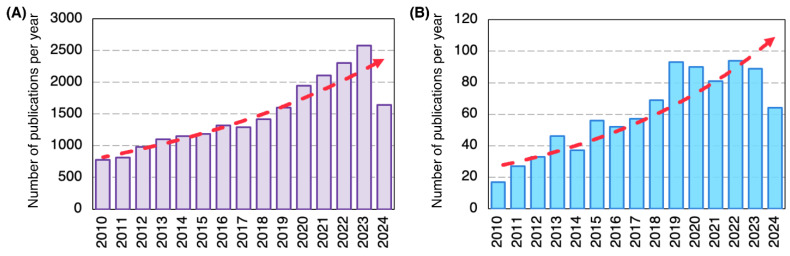
Number of scientific papers per year from 2010 to June 2024. (**A**) Publications on “Wound dressing”. (**B**) Publications on “Core-shell nanofiber”.

**Figure 2 polymers-16-02526-f002:**
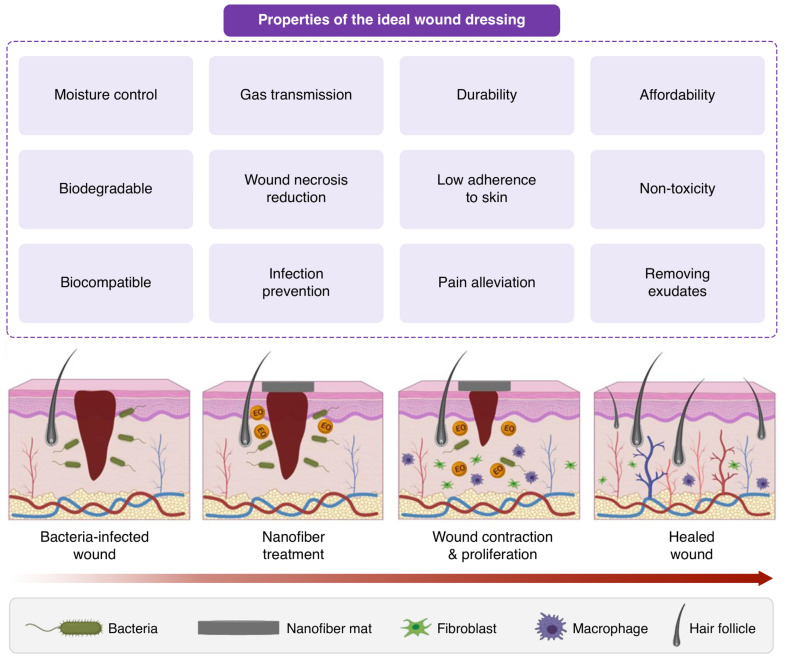
Properties of the ideal wound dressing. The bottom illustration shows the process of wound healing, containing bacteria-infected wounds, treatment with nanofibers, wound contraction and proliferation, and healed wounds. (Reproduced from ref. [33] with permission from Elsevier).

**Figure 4 polymers-16-02526-f004:**
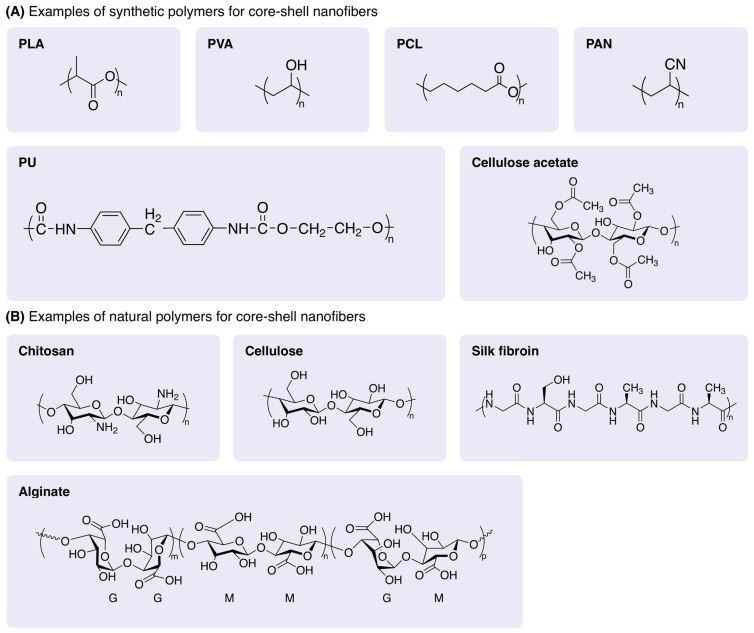
Chemical structures of the common polymers used in core-shell nanofibers. (**A**) Synthetic polymers. (**B**) Natural polymers.

**Figure 5 polymers-16-02526-f005:**
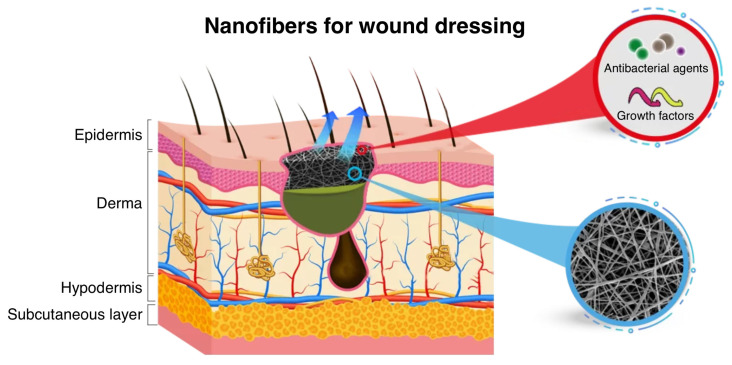
Schematic of applied nanofibers on a wound site. Layers contain epidermis, derma, hypodermis, and subcutaneous tissue. (Adapted from Invenso Technology^®^ with permission).

**Figure 6 polymers-16-02526-f006:**
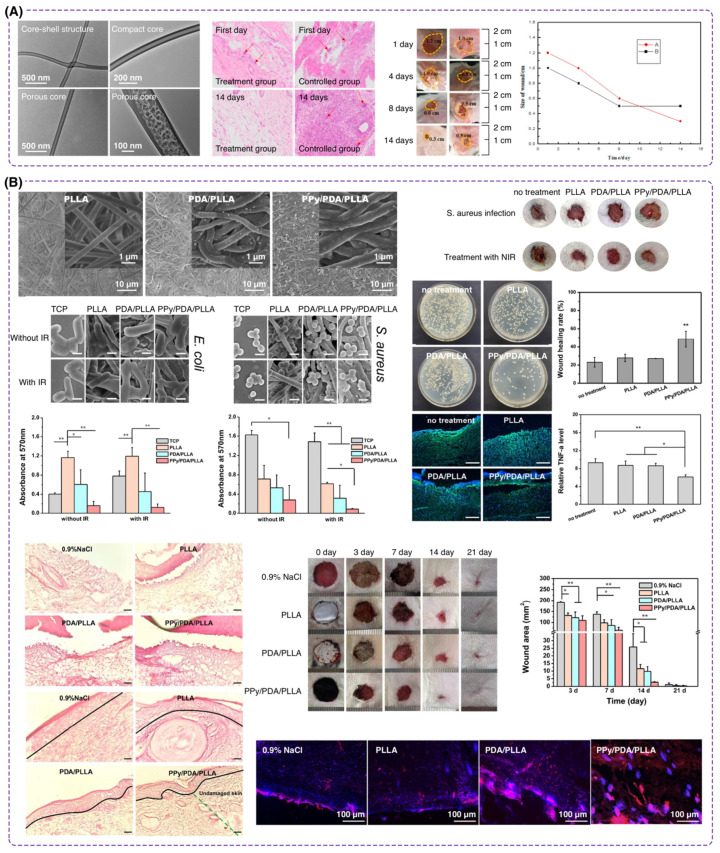
Core-shell nanofibers for wound dressing. (**A**) Various core structures in a core-shell nanofiber made of PLA and γ-PGA. H&E staining of wound tissue with images of the wound healing extension are provided. (Reproduced from ref. [113] with permission from Elsevier). (**B**) Wound healing and antibacterial activity of three-layer PPy/PDA/PLLA core-shell nanofibers. Antibacterial effects of *E. coli* and *S. aureus* cultured on different materials before and after near-infrared treatment are shown. In vivo antibacterial properties are demonstrated through infected wounds, and bacterial colony-forming units of *S. aureus* harvested from these regions. Immunohistochemical analyses of wounds on the dorsum of rats by TNF-α (green) and cell nucleus (blue) staining depict improved antibacterial properties of the dressing upon the addition of PPy. Staining of tissue sections and immunostaining of neovascularization demonstrate the evaluation of the wound repair efficacy. (Reproduced from ref. [115] with permission from Elsevier).

**Figure 8 polymers-16-02526-f008:**
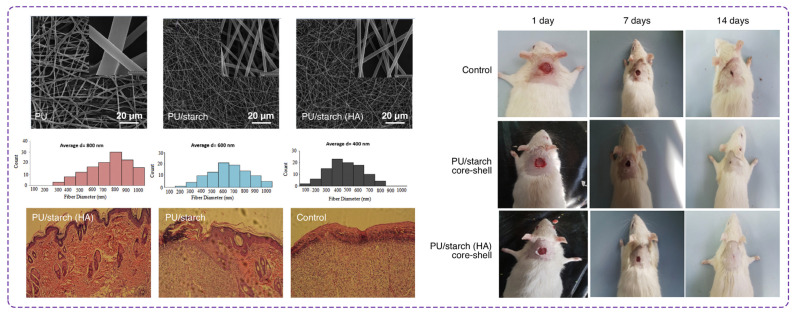
SEM images of the core-shell nanofibers made of PU, PU/starch, and PU/starch (HA). Photographs of the treated wounds with the core-shell dressings and H&E staining of wound tissue are presented. (Reproduced from ref. [142] with permission from Elsevier).

**Figure 9 polymers-16-02526-f009:**
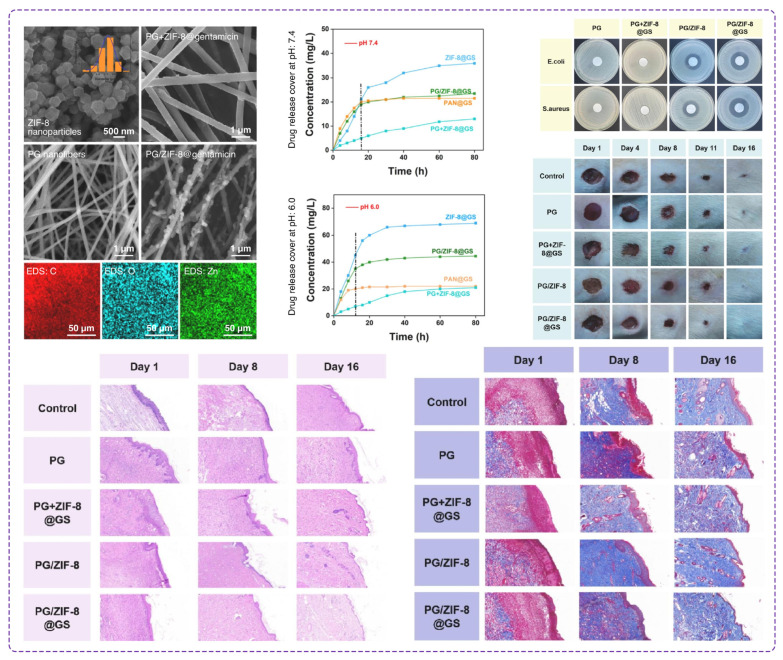
SEM images of ZIF-8 nanoparticles and PAN/gelatin (PG) nanofibers doped with ZIF-8 nanoparticles. The wound repair efficacy of the prepared nanofibers is evaluated by investigating drug release profile, antibacterial activity, in vivo wound record on rats, and H&E staining and Masson staining of wound tissue on different days. (Reproduced from ref. [154] with permission from Elsevier).

**Figure 10 polymers-16-02526-f010:**
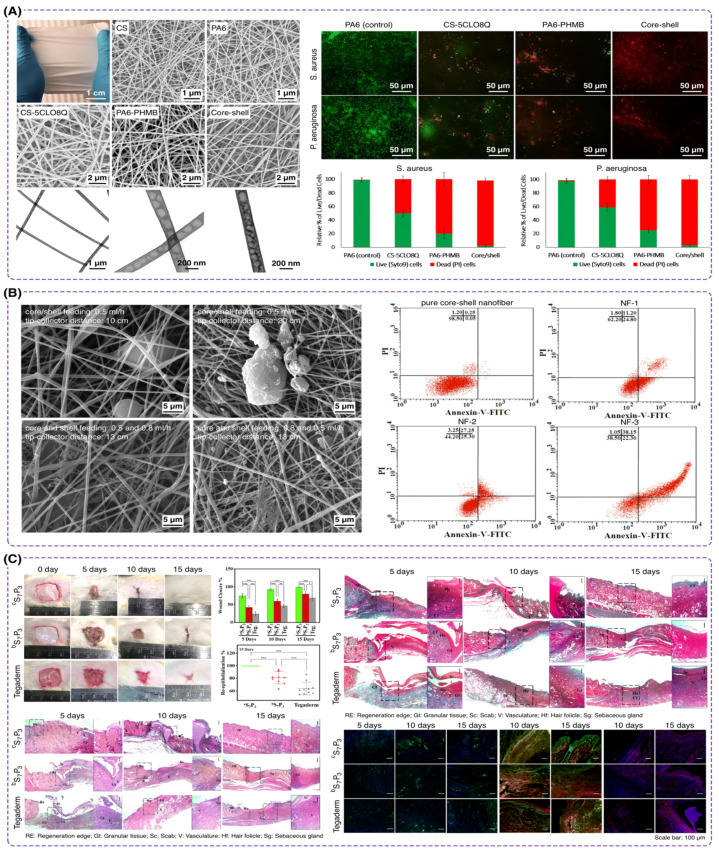
Wound dressing with core-shell nanofibers. (**A**) Image of the electrospun mat along with SEM and TEM micrographs of nylon-6/chitosan core-shell nanofibers. Fluorescence microscopy assessment via live/dead BacLight Syto9/Propidium iodide (PI) bacterial viability assay. (Reproduced from ref. [165] with permission from Springer). (**B**) SEM images of the chitosan/PCL nanofibers fabricated with various synthesis parameters. Flow cytometry analyses of MCF-7 cells treated with dissimilar nanofibers are presented. (Reproduced from ref. [166] with permission from Elsevier). (**C**) In vivo wound healing study with high fibroin-loaded silk/PCL nanofibers. Wound repair evaluation is investigated through H&E staining of wound tissue with histological microscopic pictures, MT stain microscopic images for collagen deposition, and fluorescence images. (Reproduced from ref. [167] with permission from the Royal Society of Chemistry).

**Figure 11 polymers-16-02526-f011:**
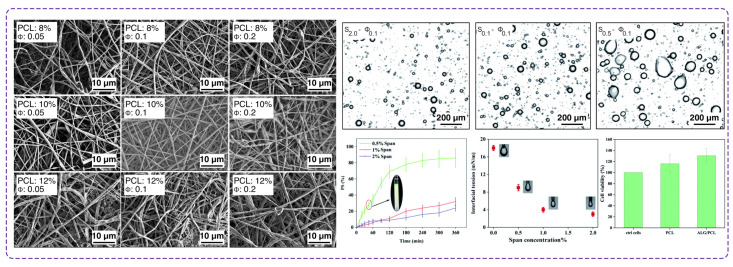
Emulsion electrospinning of sodium alginate/PCL nanofibers in water/oil emulsion. Obtained cell viability of normal human dermal fibroblasts exposed to different membranes is shown. (Reproduced from ref. [183] with permission from The Royal Society of Chemistry).

**Figure 12 polymers-16-02526-f012:**
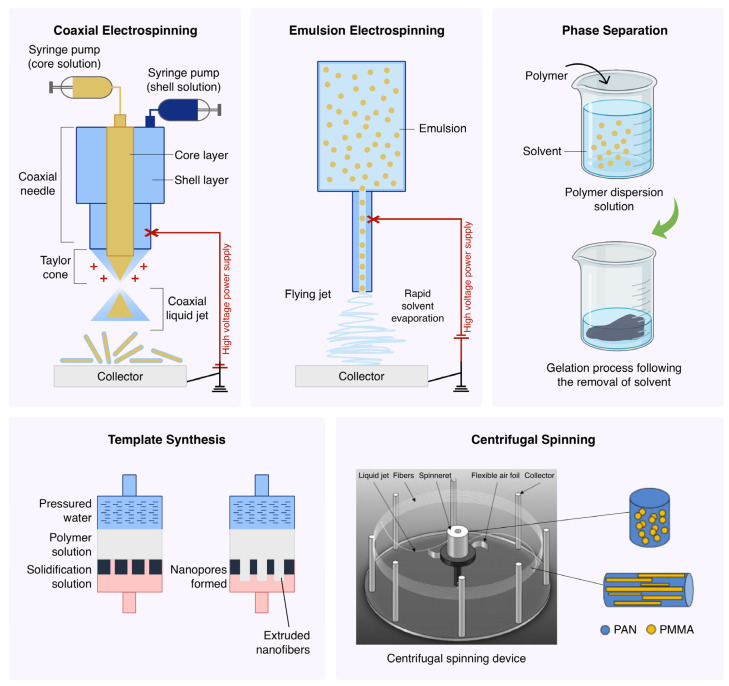
Common synthesis methods for fabricating core-shell nanofibers, embodying coaxial electrospinning, emulsion electrospinning, phase separation, template synthesis, and centrifugal spinning. (The image of the centrifugal spinning device was adapted from ref. [243] with permission from Elsevier.).

**Table 1 polymers-16-02526-t001:** Recently published review papers in the field of core-shell nanofibers for wound dressing applications.

Source Title	Year	Aim	References
Electrospun nanofiber membranes with various structures for wound dressing	2023	Reviewed the development and application of electrospun nanofiber membranes for wound dressings, focusing on structural design and the incorporation of therapeutic factors.	[23]
State-of-the-art review of advanced electrospun nanofiber composites for enhanced wound healing	2023	Presented the advancement in electrospun nanofiber composites for wound healing with an emphasis on morphologies and methods.	[24]
Hyaluronic acid and chitosan-based electrospun wound dressings: Problems and solutions	2022	Centered on the factors affecting the electrospinning of hyaluronic acid and chitosan for wound dressing applications including their biological roles and mechanisms.	[25]
In vitro and in vivo advancement of multifunctional electrospun nanofiber scaffolds in wound healing applications: Innovative nanofiber designs, stem cell approaches, and future perspectives	2022	Discussed nanofiber geometries with their potentials and stem cell approaches for electrospun nanofiber scaffolds.	[26]
Nature-derived and synthetic additives to poly(ɛ-caprolactone) nanofibrous systems for biomedicine: An updated overview	2022	Reviewed recent advancements in PCL nanofibers for biomedical and tissue engineering applications to improve their properties since 2017.	[27]
A review on biopolymer-derived electrospun nanofibers for biomedical and antiviral applications	2022	Reviewed the efficiency and optimization of electrospinning method in fabrication of multifunctional nanofibers for biomedical and tissue regeneration applications.	[28]
Review on nanoparticles and nanostructured materials: Bioimaging, biosensing, drug delivery, tissue engineering, antimicrobial, and agro-food applications	2022	Presented synthesis methods and applications of various nanomaterials in agricultural and biomedical related fields.	[29]
Natural protein-based electrospun nanofibers for advanced healthcare applications: Progress and challenges	2022	Highlighted the advancement and challenges in natural protein-based electrospun nanofibers for wound dressing, tissue engineering, and drug delivery applications.	[30]
Nanofibers for biomedical and healthcare applications	2019	Focused on the recent reports on fabrication, scaling-up challenges, and application of electrospun nanofibers, featuring their potential in drug delivery, wound healing, and tissue engineering applications.	[31]
Cellulose acetate electrospun nanofibers for drug delivery systems: Applications and recent advances	2018	Reviewed the methods, applications, and opportunities of cellulose acetate electrospun nanofibers in drug delivery systems.	[32]

**Table 2 polymers-16-02526-t002:** Common nanomaterials with antibacterial activity used in core-shell nanofibers.

Nanomaterial	Abbreviation	Structure	Dimension (nm)	Surface Area (m^2^/g)	Mechanism of Toxicity
Silver nanoparticles	AgNPs	Spherical	10–20	20–50	ROS, Ag^+^ release, inflammatory responses, genotoxicity, mitochondrial dysfunction
Zinc oxide	ZnO	Hexagonal	10–30	10–50	ROS, Zn^2+^ ion release, inflammatory responses, genotoxicity
Graphene oxide	GO	Sheet	100–200	2630	ROS, physical cell membrane disruption, adsorption of biomolecules and starvation
Copper oxide	CuO	Rod-like	10–50	20–80	ROS, Cu^2+^ release, inflammatory responses, binding to bacterial proteins and enzymes
Titanium oxide	TiO_2_	Rod-like and anatase	5–50	50–200	ROS, photocatalytic activity, physical cell membrane disruption, adsorption biomolecules, genotoxicity

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
