# Peer review of "Wound Dressing with Electrospun Core-Shell Nanofibers: From Material Selection to Synthesis"

_polymers, 2024, doi:10.3390/polym16172526_

Round 1

Reviewer 1 Report

Comments and Suggestions for Authors

The manuscript presents a very interesting review on the topic of the electrospun core-shell nanofibers for wound healing applications.  The topic is very interesting and the way the authors decided to present it (i.e. the structure of the review, the division in sections and subsections) is appropriate for such a kind of work.

The topic is very extended in literature and such a review study cannot be exhaustive. Having this in mind, the authors managed to cite important literature contributions and included a large number of studies.

Some issues that should be addressed before publishing are:

- Some language issues should be addressed before publishing.

- Polyvinyl alcohol is reported in section 4.1.1 under the title “Polyesters”. However, PVA is not a polyester. Probably the title of this section should be revised (maybe it should be Synthetic Polymers?).

- Also, chitosan is not a natural polymer (section 4.1.2). It is a semi-synthetic polymer and it is a derivative of chitin. Chitin is the natural polymer. Thus, the authors should also consider revising the title of section 4.1.2.

- Section 4.1.2. The text for Silk fibroin is written twice. Please remove the duplicated part.

- It is the Reviewer’s opinion that cellulose acetate should be also mentioned in section 4.1.2

Comments on the Quality of English Language

Moderate editing of English language required.

Author Response

Reviewer 1:

Corresponded revisions will be highlighted in Blue in the manuscript.

The manuscript presents a very interesting review on the topic of the electrospun core-shell nanofibers for wound healing applications.  The topic is very interesting and the way the authors decided to present it (i.e. the structure of the review, the division in sections and subsections) is appropriate for such a kind of work.

The topic is very extended in literature and such a review study cannot be exhaustive. Having this in mind, the authors managed to cite important literature contributions and included a large number of studies.

Some issues that should be addressed before publishing are:

- Some language issues should be addressed before publishing.

Response: We are grateful for the suggestion of the respected reviewer. The presented revision has undergone a thorough assessment, ensuring the language and choice of words are clear and decipherable enough for readers.

- Polyvinyl alcohol is reported in section 4.1.1 under the title “Polyesters”. However, PVA is not a polyester. Probably the title of this section should be revised (maybe it should be Synthetic Polymers?).

Response: We appreciate the perceptive comment of the reviewer. The subheading was revised per request as we found their suggestions precise and useful.

- Also, chitosan is not a natural polymer (section 4.1.2). It is a semi-synthetic polymer and it is a derivative of chitin. Chitin is the natural polymer. Thus, the authors should also consider revising the title of section 4.1.2.

Response: We are grateful for the instructive feedback of the reviewer. Although we also agree with your comment as we stated in the manuscript line 681-682, literature has highly referred to chitosan as a natural polymer (i.e., biopolymer), rationalized by being a derivative of chitin. Due to this fact, we have classified it as a natural polymer. Some of the references are provided below. To be consistent with the previous and ongoing work, it would be therefore beneficial to maintain the same dialogue.

https://doi.org/10.1002/mabi.200300019

https://doi.org/10.3390/app5041272

https://doi.org/10.3390/molecules25173981

Research paper available on ResearchGate

https://doi.org/10.1021/acsabm.1c00078

- Section 4.1.2. The text for Silk fibroin is written twice. Please remove the duplicated part.

Response: We are thankful for the perceptive comment of the respected reviewer. A revision was accordingly employed to the manuscript.

- It is the Reviewer’s opinion that cellulose acetate should be also mentioned in section 4.1.2

Response: We appreciate the kind suggestion of the respected reviewer. We have included a survey on cellulose acetate in wound dressings as requested.

Reviewer 2 Report

Comments and Suggestions for Authors

1.     Original submission

1.1.Recommendation:  

Major revision.

2.     Comments to author:

Journal: Polymers (MDPI)

Title: Wound dressing with electrospun core-shell nanofibers: From material selection to synthesis

Overview and general recommendation:

There's a certain novelty value in this work. I recommend its publication, but some major revisions should be addressed before reconsideration. Several suggestions have been given below:

2.1. Major concerns:

(1)  The abstract should be revised and improved.

(2)  In the last part of the Introduction, the difference of this study from the studies in the literature should be added. (Is there a review on this subject before? If so, what is the difference between this study and other articles?)

(3)  Recently, some good articles are published to fabricate electrospun nanofibers. I advise the authors to show more respect to the works discussing the importance of electrospun nanocomposites as eco-friendly materials. As far as I know, there are some quite representative papers published recently. I could see that the paper selectively cited few works in this aspect, but failed to mention other contributions such as: Chemical Engineering Journal, 2024, 493, 152422; Mater. Chem. B. 2020, 8, 3701-3732; and Chemical Engineering Journal, 2023, 452, 139060, which are very interesting.

(4)  The authors should emphasize the relationship between composite structure and properties along with the underlying mechanisms.

(5)  I would like to see the advantages and disadvantages of the preparation techniques of fibers comparatively in a Table. For example, what is the easiest method and least costly method?

(6)  The authors in Conclusions should make a comparison with promising Wound dressing with electrospun core-shell nanofiber materials

(7)  The caption of Figure 6 should be revised.

(8)  The "Conclusion" section should be revised and improved..

Other comments:

Q1: The subject addressed in this article is worthy of investigation: Agree

Q2: The information presented is new: Agree

Q3: The conclusions are supported by the data: Neutral

Q4: The manuscript is appropriate for the journal: Agree

Q5: Organization of the manuscript is appropriate: Agree

Q6: Figures, tables and supplementary data are appropriate: Agree

Author Response

Reviewer 2:

Corresponded revisions will be highlighted in Green in the manuscript.

Overview and general recommendation:

There's a certain novelty value in this work. I recommend its publication, but some major revisions should be addressed before reconsideration. Several suggestions have been given below:

2.1. Major concerns:

(1)  The abstract should be revised and improved.

Response: We are grateful for the comment of the respected reviewer. The presented revision has undergone a thorough assessment, ensuring the language and choice of words are clear and decipherable enough for readers.

(2)  In the last part of the Introduction, the difference of this study from the studies in the literature should be added. (Is there a review on this subject before? If so, what is the difference between this study and other articles?)

Response: We appreciate the feedback of the respected reviewer. As stated in the last paragraph of the introduction (lines 83-89) along with the second section (lines 122-127), there are no reports on wound dressing using electrospun core-shell nanofibers. The structure as well as the novelty of the current work has been addressed fully in section 2 featuring Table 1.

(3)  Recently, some good articles are published to fabricate electrospun nanofibers. I advise the authors to show more respect to the works discussing the importance of electrospun nanocomposites as eco-friendly materials. As far as I know, there are some quite representative papers published recently. I could see that the paper selectively cited few works in this aspect, but failed to mention other contributions such as: Chemical Engineering Journal, 2024, 493, 152422; Mater. Chem. B. 2020, 8, 3701-3732; and Chemical Engineering Journal, 2023, 452, 139060, which are very interesting.

Response: We are happy to hear such an instructive comment from the respected reviewer. The suggested references have been cited in the revised manuscript.

(4)  The authors should emphasize the relationship between composite structure and properties along with the underlying mechanisms.

Response: We thank the respected reviewer for the perceptive comments. We are aware of the importance and role of the inclusion of micro and nanomaterials in nanofibers for obtaining higher or controlled performances. Some of the mechanisms involved in the improved properties come from interfacial bonding, microphase structure, orientation of the constituents, processing, and the dispersion and/or distribution of the reinforcement agents.

First, the quality of bonding between the components influences properties like adhesion strength, fracture toughness, and resistance to delamination. If the interfacial bonding is strong enough, it can improve the mechanical properties and durability of the prepared structure. The control over phase morphology also optimizes properties as another mechanism influenced by the composite structure. We should note that because the system is a mixture of polymer and a secondary element to form a composite, the orientation and alignment of the reinforcements play a pivotal role in the ultimate properties. Impacted by localization as well as orientation/alignment of fibers, particles, or other reinforcing elements within the matrix, a set of properties including anisotropy, stiffness, and impact resistance are subject to change. For multiphase systems, the dispersion and distribution of the functional element are vital due to the correlated outcome on the mechanical properties of the composite structure. In fact, load-bearing and deformation resistance significantly improves upon adding reinforcement if such a uniform structure is obtained. Regardless of the material’s aspect, the processing method not only can improve the overall properties of polymeric structures but also be highly effective for polymer/micro- or nanomaterials. The selection of proper processing techniques results in achieving the desired properties and uniformity within the composite.

Under each subheading, the technical discussion has been considered with respect to the corresponding impacts on wound healing processes, as well as the physico-mechanical properties of core-shell nanofibers.

(5)  I would like to see the advantages and disadvantages of the preparation techniques of fibers comparatively in a Table. For example, what is the easiest method and least costly method?

Response: We are grateful for the constructive feedback of the respected reviewer. We have provided such a comparison in Table 3.

(6)  The authors in Conclusions should make a comparison with promising Wound dressing with electrospun core-shell nanofiber materials.

Response: We are thankful for the kind suggestion of the respected reviewer. The precise control over the bioactive agents’ release is granted from the individual material compartment with self-reliant properties in core-shell nanofibers. Because of such structures, core-shell nanofibers are competent in moisture management, promoting faster healing and preventing complications. Also, the excellent mechanical strength of core-shell nanofibers enhances their durability and resilience in wound dressing applications, supported by the provided survey in the manuscript. Recent work further showed the customability of core-shell nanofibers to mimic the extracellular matrix, which is essential for cellular adhesion and proliferation. In terms of processing and application, core-shell nanofibers allow for a high degree of customization in design, enabling the incorporation of diverse bioactive agents and materials to address specific wound healing requirements. This versatility makes them adaptable to a wide range of wound types and patient needs, notably when several synthesis methods are recognized to manufacture the nanofibers. The conclusion includes a comprehensive statement of the state-of-the-art core-shell nanofibers for wound dressing with respect to the importance of material selection and synthesis methods. We have disclosed our discussion by mentioning the ongoing direction of this field to showcase the potential of core-shell nanofibers for wound dressings.

(7)  The caption of Figure 6 should be revised.

Response: We appreciate the perceptive comment of the respected reviewer. The revision was provided accordingly in the manuscript.

(8)  The "Conclusion" section should be revised and improved.

Response: We are grateful for the suggestion of the respected reviewer. The presented revision has undergone a thorough assessment, ensuring the language and choice of words are clear and decipherable enough for readers.
